# GoalRank: Group-Relative Optimization for a Large Ranking Model

**Kaike Zhang**[1,2*†]   **Xiaobei Wang**[1*]   **Shuchang Liu**[1]   **Hailan Yang**[1]   **Xiang Li**[1]
**Lantao Hu**[1]          **Han Li**[1]              **Qi Cao**[2]            **Fei Sun**[2]           **Kun Gai**[1]

[1] Kuaishou Technology, Beijing, China
[2] University of Chinese Academy of Sciences, Beijing, China
{kaikezhang99, caoqi92seven, ofey.sunfei}@gmail.com
{wangxiaobei03, liushuchang, yanghailan}@kuaishou.com
{lixiang44, hulantao, lihan08, yuyue06}@kuaishou.com

## Abstract

Mainstream ranking approaches typically follow a Generator–Evaluator two-stage paradigm, where a generator produces candidate lists and an evaluator selects the best one. Recent work has attempted to enhance performance by expanding the number of candidate lists, for example, through multi-generator settings. However, ranking involves selecting a recommendation list from a combinatorially large space, simply enlarging the candidate set remains ineffective, and performance gains quickly saturate. At the same time, recent advances in large recommendation models have shown that end-to-end one-stage models can achieve promising performance with the expectation of scaling laws. Motivated by this, we revisit ranking from a generator-only one-stage perspective. We theoretically prove that, for any (finite Multi-)Generator–Evaluator model, there always exists a generator-only model that achieves strictly smaller approximation error to the optimal ranking policy, while also enjoying a scaling law as its size increases. Building on this result, we derive an evidence upper bound of the one-stage optimization objective, from which we find that one can leverage a reward model trained on real user feedback to construct a reference policy in a *group-relative* manner. This reference policy serves as a practical surrogate of the optimal policy, enabling effective training of a large generator-only ranker. Based on these insights, we propose **GoalRank**, a generator-only ranking framework. Extensive offline experiments on public benchmarks and large-scale online A/B tests demonstrate that **GoalRank** consistently outperforms state-of-the-art methods.

## 1 Introduction

Recommender systems are indispensable for coping with the exponential growth of online content (Gomez-Uribe & Hunt, 2015). Industrial platforms typically adopt a multi-stage pipeline, comprising retrieval (He et al., 2020; Zhang et al., 2024) and ranking (Yu et al., 2019; Liu et al., 2023; Zhang et al., 2025b). The ranking stage is particularly critical, as it determines the final sequence of items shown to users and has a major impact on both user satisfaction and platform revenue.

Formally, the ranking task can be defined as an $N{\to}L$ list-generation problem: given $N$ candidates from the preceding stage, the model outputs an ordered list of length $L$. The search space is the set of length-$L$ permutations, $\mathrm{P}(N, L) = \frac{N!}{(N-L)!}$, which makes exhaustive enumeration intractable for large $N$. Early approaches adopt a **one-stage single generator** that directly produces recommendation lists by scoring items and arranging them greedily (Zhuang et al., 2018; Ai et al., 2018; Pei et al., 2019a; Gong et al., 2022; Liu et al., 2023), as illustrated in Figure 1(a). However, this greedy strategy only models the item interdependencies in the candidate set (of size $N$) but not in the output list (of size $L$), often resulting in suboptimal rankings.

---

*Equal contribution.
†Corresponding author.

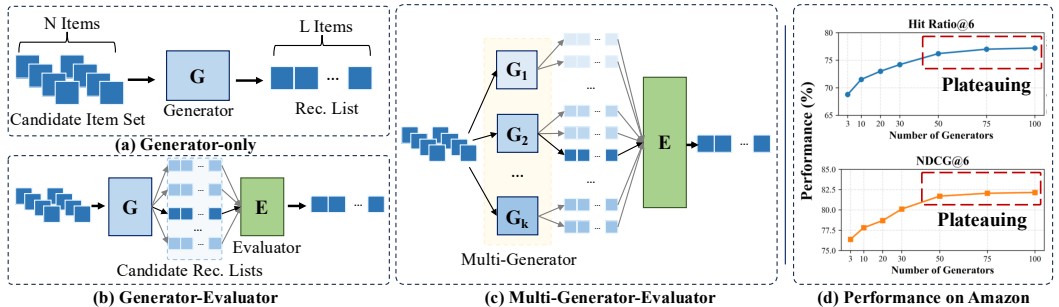

Figure 1: Illustration of different ranking paradigms: (a) Generator-only; (b) Generator–Evaluator; (c) Multi-Generator–Evaluator; and (d) Performance trend with increasing number of generators.

To address this limitation, subsequent studies propose a **two-stage Generator–Evaluator** paradigm (Shi et al., 2023; Xi et al., 2024; Lin et al., 2024; Ren et al., 2024b; Zhang et al., 2025b) (Figure 1b): a generator first proposes multiple candidate lists, and an evaluator then selects the best one according to an estimated list-wise value. To mitigate the risk that generators produce only locally optimal candidates, later works introduce **multi-generator** settings (Figure 1c), thereby increasing both the number and diversity of candidate lists. In practice, however, simply scaling the number of candidates or generators yields diminishing returns, with performance gains quickly plateauing (Figure 1d).

Meanwhile, advances in **end-to-end, one-stage** large recommendation models suggest that a single sufficiently expressive model can subsume multi-stage pipelines, avoid cross-stage inconsistencies, and exhibit favorable scaling behavior (Zhai et al., 2024; Deng et al., 2025). These findings indicate that the two-stage Generator–Evaluator paradigm may not be indispensable for achieving high-quality ranking. Motivated by this, we revisit the **generator-only** paradigm and ask: can a larger, more powerful one-stage ranker directly produce high-quality lists without relying on an external evaluator? Formally, let $\pi^*$ denote the optimal ranking policy. We focus on two central questions:

i *For any (finite Multi-)Generator–Evaluator system, does there exist a single generator-only model whose policy achieves a strictly smaller approximation error with respect to $\pi^*$?*

ii *If such a model exists, how can it be trained effectively to realize this approximation advantage in practice?*

To answer these questions, we analyze (in Section 3.1) the approximation error between the policy space induced by a finite set of (Multi-)Generator–Evaluator models and the optimal ranking policy $\pi^*$. This analysis proves the existence of a generator-only model that can achieve a strictly smaller approximation error. Moreover, we show that as the size of this generator-only ranking model increases, its approximation error with respect to $\pi^*$ decreases accordingly. Building on these theoretical insights, we then turn to the practical challenge of how to train such a one-stage ranking model. By deriving an evidence upper bound of the existing optimization objective, we find that one can leverage a reward model trained on real user feedback to construct a reference policy in a *group-relative* manner, which serves as a surrogate for $\pi^*$. This enables us to train a large generator-only ranking model effectively. Based on this idea, we propose a new training framework, **GoalRank** (**G**roup-Relative **O**ptimiz**A**tion for a **L**arge **Rank**er). We validate the effectiveness of **GoalRank** on public benchmarks as well as through large-scale online A/B tests, showing substantial improvements over state-of-the-art baselines and clear evidence of scaling laws.

The main contributions of this work can be summarized as follows:

- **Theoretical foundation.** We prove that for any (finite Multi-)Generator–Evaluator family, there always exists a generator-only model that achieves a strictly smaller approximation error to the optimal ranking policy, and that this error decreases as model size increases (scaling law).

- **Optimization principle.** We introduce the Group-Relative optimization principle, which provides a tractable and effective criterion for training large generator-only ranking models.

- **Model and validation.** We instantiate these ideas in **GoalRank**, a generator-only large ranker trained under the proposed principle. Extensive offline experiments and online A/B tests demon-

strate consistent improvements over strong baselines and reveal clear scaling laws with respect to model capacity.

## 2 RELATED WORK AND PRELIMINARIES

The ranking task in recommender systems can be formulated as an $N{\to}L$ list-generation problem: given $N$ candidate items, the model outputs an ordered list of length $L$. Let $\mathcal{U}$ and $\mathcal{V}$ denote the user and item sets. For each user $u$, the candidate set is $\mathcal{V}_u \subseteq \mathcal{V}$ with $|\mathcal{V}_u| = N$. The generation space is

$$\mathcal{L}_u = \left\{ (v_1, \ldots, v_L) \in \mathcal{V}_u^L : v_i \neq v_j\ (i \neq j) \right\}, \qquad |\mathcal{L}_u| = \mathrm{P}(N, L) = \tfrac{N!}{(N-L)!}.$$

Since $\mathrm{P}(N, L)$ is exponentially large, existing works build approximate ranking paradigms to explore this space.

**Single-Stage Generator-Only Models.** A straightforward solution is to use a single generator $G$ that scores items and constructs a list greedily (Zhuang et al., 2018; Ai et al., 2018; Pei et al., 2019a; Gong et al., 2022; Liu et al., 2023; Feng et al., 2021b; Xi et al., 2022; Pei et al., 2019b). Classic models include DLCM (Ai et al., 2018) and PRM (Pei et al., 2019a), which refine item scores using local listwise features. Formally:

$$l_u^* = G(\mathcal{X}_u, \mathcal{V}_u).$$

Although efficient, these methods typically underexplore inter-item dependencies and may not remain consistent with the conditioning of the original candidate set.

**Two-Stage Generator–Evaluator Paradigm.** To better address the combinatorial $N{\to}L$ search space, recent work adopts a two-stage (multi-)Generator–Evaluator (G–E) framework (Chen et al., 2022; Shi et al., 2023; Xi et al., 2024; Lin et al., 2024; Ren et al., 2024b; Wang et al., 2025b). A generator produces multiple candidate lists, and an evaluator scores them to select the best one:

$$l_u^* = \arg \max_{l \in \mathcal{L}_{u,k}} E(\mathcal{X}_u, l), \qquad \mathcal{L}_{u,k} = \left\{ G_i(\mathcal{X}_u, \mathcal{V}_u) \mid i = 1, \ldots, k \right\}.$$

The special case $k{=}1$ reduces to a single-generator–evaluator model. Multi-generator extensions ($k > 1$) aim to enlarge the proposal space (Yang et al., 2025), but empirical gains saturate rapidly as $k$ grows. This diminishing return suggests that merely increasing the number of generators is inefficient and highlights the need for fundamentally stronger listwise modeling.

**Other Directions.** Other concurrent efforts incorporate large language models (Ren et al., 2024a; Gao et al., 2024; Wu et al., 2024; Gao et al., 2025; Ren et al., 2025; Liu et al., 2025) or reinforcement learning (Feng et al., 2021c; Wang et al., 2024; 2025c; Wei et al., 2020). LLM-based methods leverage textual side information, while RL-based approaches decompose the listwise value function to align rankings with user utility.

## 3 METHODOLOGY

In this section, we address the research questions raised in Section 1, namely: (i) can the generator-only paradigm outperform the (Multi-)Generator–Evaluator paradigm, and (ii) if so, how can such a generator-only ranking model be effectively learned? Building on the insights gained from these analyses, we then propose a new generator-only large ranker framework, **GoalRank**, which leverages group-relative optimization to approximate the optimal ranking policy.

### 3.1 CAN THE GENERATOR-ONLY PARADIGM PERFORM BETTER?

To assess the feasibility of a single-stage large ranking model, we first ask whether a sufficiently large generator-only model can match or even exceed the expressive power of the widely used two-stage (Multi-)Generator-Evaluator pipeline. Formally, suppose there exists an optimal ranking policy $\pi^*$. We compare the best attainable approximation error of (i) a $k$-mixture of small generators combined with an evaluator, and (ii) a single larger generator. To make this comparison precise, we begin by defining a capacity-restricted generator class.

**Definition 1** (($\alpha, \beta$)-bounded generator class). *Given maximum generator width $\alpha$ and depth $\beta$, the ($\alpha, \beta$)-bounded generator class is defined as*

$$\mathcal{G}_m(\alpha, \beta) := \left\{ g_m \mid W(g_m) \leq \alpha, \ D(g_m) \leq \beta \right\},$$

*where $g_m$ denotes a generator, and $W(\cdot)$ and $D(\cdot)$ measure width- and depth-type complexities, respectively.*

Then, the evaluator can be regarded as operating over a low-dimensional probability simplex, which determines how multiple small generators jointly influence the final ranking policy.

**Definition 2** ($k$-mixture ($\alpha, \beta$)-bounded policy space). *The policy space induced by $\mathcal{G}_m(\alpha, \beta)$ is*

$$\mathcal{F}_m(\alpha, \beta) := \left\{ \mathrm{softmax} \circ g_m \mid g_m \in \mathcal{G}_m(\alpha, \beta) \right\},$$

*which contains all policies realizable by a single generator in $\mathcal{G}_m(\alpha, \beta)$ with a softmax output layer. Given $k$ generators in $\mathcal{G}_m(\alpha, \beta)$ and an evaluator, the corresponding $k$-mixture ($\alpha, \beta$)-bounded policy space is*

$$\mathcal{C}_m^k(\alpha, \beta) := \left\{ \sum_{i=1}^k \omega_i \pi_i \ \middle| \ \boldsymbol{\omega} \in \Delta^{k-1}, \ \pi_i \in \mathcal{F}_m(\alpha, \beta) \right\},$$

*where $\Delta^{k-1}$ is the ($k-1$)-dimensional probability simplex and $\boldsymbol{\omega} = (\omega_1, \ldots, \omega_k)$ satisfies $\sum_{i=1}^k \omega_i = 1$ and $\omega_i \geq 0$.*

In Definition 2 we adopt soft mixture weights $\boldsymbol{\omega}$. In practice, the evaluator often implements (or approximates) a hard selection (one-hot $\boldsymbol{\omega}$). Thus, $\mathcal{C}_m^k(\alpha, \beta)$ strictly contains the policy class realized by hard selection, which can both simplifies subsequent derivations and strengthens Theorem 1. Then, to evaluate how well a policy space approximates the optimal ranking policy $\pi^*$, we use the following notion.

**Definition 3** (Approximation distance (KL error)). *Let $\pi^*$ be a target policy and $\mathcal{F}$ be a policy space. The approximation distance from $\mathcal{F}$ to $\pi^*$ is*

$$\mathcal{E}(\mathcal{F}) := \inf_{\pi \in \mathcal{F}} \mathrm{KL}(\pi^* \| \pi), \qquad \mathrm{KL}(\pi^* \| \pi) = \sum_{l \in \mathcal{L}} \pi^*(l) \log \frac{\pi^*(l)}{\pi(l)}.$$

*where $\mathcal{L}$ denotes the finite space of candidate lists considered by the ranker.*

With these definitions in place, we can now state our main result.

**Theorem 1.** *Given $\alpha, \beta > 0$ and any $k \in \mathbb{N}_{>0}$. For the $k$-mixture policy space $\mathcal{C}_k^m(\alpha, \beta)$ in Definition 2, there exists a class of larger generators*

$$\mathcal{G}_M(\alpha, \beta, n) := \left\{ g_M \mid W(g_M) \geq k\alpha + n, \ D(g_M) \geq \beta \right\}, \qquad n \in \mathbb{N}_{>0},$$

*with associated policy space*

$$\mathcal{F}_M(\alpha, \beta, n) := \left\{ \mathrm{softmax} \circ g_M \mid g_M \in \mathcal{G}_M(\alpha, \beta, n) \right\},$$

*such that*

$$\mathcal{E}\big(\mathcal{F}_M(\alpha, \beta, n)\big) < \mathcal{E}\big(\mathcal{C}_m^k(\alpha, \beta)\big), \qquad \lim_{n \to \infty} \mathcal{E}\big(\mathcal{F}_M(\alpha, \beta, n)\big) = 0.$$

Proofs and technical details are deferred to Appendix A. Theorem 1 shows that for any two-stage ranking mixing $k$ small generators, there exists a sufficiently large one-stage generator-only ranking model whose induced policy space achieves a strictly smaller approximation error to $\pi^*$. Moreover, as the size of this generator increases (i.e., as $n$ grows), the approximation error can be driven arbitrarily close to zero. We remark that Theorem 1 is stated in terms of width scaling; the same conclusion holds under depth scaling, with proofs provided in Appendix A.

### 3.2 How can generator-only ranking model be effectively learned?

According to Theorem 1, our goal is to train a larger generator-only ranking model that can achieve a closer approximation to the optimal ranking policy $\pi^*$. Suppose we have access to an ideal reward

model $r^*(l)$ that provides an unbiased estimate of the user feedback for any candidate list $l \in \mathcal{L}_u$ of user $u^1$. We define the entropy-regularized oracle policy as

$$\pi^* := \arg\max_\pi \left\{ \mathbb{E}_{l \sim \pi}[r^*(l)] + \tau \mathcal{H}(\pi) \right\}, \tag{1}$$

where $\mathcal{H}(\pi)$ denotes the entropy of the policy, introduced as a regularization term to avoid greedy instability and to encourage exploration, and $\tau > 0$ controls the strength of entropy regularization. Optimizing Equation 1 yields the Boltzmann distribution

$$\pi^*(l) = \frac{\exp(r^*(l)/\tau)}{Z}, \qquad Z = \sum_{l'} \exp(r^*(l')/\tau). \tag{2}$$

Moreover, the objective in Equation 1 can be equivalently rewritten as

$$\mathbb{E}_{l \sim \pi}[r^*(l)] + \tau \mathcal{H}(\pi) = \tau \sum_l \pi(l) \left( \log \frac{\exp(r^*(l)/\tau)}{Z} - \log \pi(l) + \log Z \right).$$

$$\text{Thus,} \qquad \tau \log Z = \sup_\pi \left\{ \mathbb{E}_{l \sim \pi}[r^*(l)] + \tau \mathcal{H}(\pi) \right\},$$

and the supremum is attained if and only if $\mathrm{KL}(\pi \| \pi^*) = 0$. In this case, the objective achieves its maximum. Therefore, optimizing $\pi$ is equivalent to minimizing the KL divergence to $\pi^*$.

In practice, however, the ideal reward model $r^*(l)$ is inaccessible. We therefore consider a potentially biased reward model $\hat{r}(l) = r^*(l) + b(l)$, where $b(l)$ denotes the bias. Intuitively, the smaller the bias, the more reliable the reward model. When considering a list group $\mathcal{B}$, if the reward gaps among lists are sufficiently large, the contribution of $r^*(l)$ dominates the bias $b(l)$, such that the (partial) order over $\mathcal{B}$ is approximately preserved. Formally, given a threshold $\sigma^* > 0$, if

$$\max_{l_i, l_j \in \mathcal{B}} \left| \hat{r}(l_i) - \hat{r}(l_j) \right| > \sigma^*, \tag{3}$$

we can exploit this order-invariance to construct a **reference policy in a group-relative manner**:

$$\pi^{\mathrm{ref}}(l \mid \mathcal{B}) = \frac{\exp\left( (\hat{r}(l) - \bar{r}_\mathcal{B})/\sigma_\mathcal{B} \right)}{\sum_{l'} \exp\left( (\hat{r}(l') - \bar{r}_\mathcal{B})/\sigma_\mathcal{B} \right)}, \tag{4}$$

where $\bar{r}_\mathcal{B}$ and $\sigma_\mathcal{B}$ are the mean and standard deviation of $\hat{r}$ over $\mathcal{B}$. We then train a parametric policy $\pi_\theta$ to align with $\pi^{\mathrm{ref}}$ by minimizing the cross-entropy (equivalently, the KL divergence up to a constant): Finally, the training objective can be expressed as

$$\mathcal{L}(\pi_\theta) = -\mathbb{E}_{\mathcal{B} \sim \mathcal{D}} \left[ \sum_{l \in \mathcal{B}} \pi^{\mathrm{ref}}(l \mid \mathcal{B}) \log \pi_\theta(l) \right]. \tag{5}$$

This objective provides a tractable surrogate for minimizing $\mathrm{KL}(\pi_\theta \| \pi^*)$ using only $\hat{r}$ and group-relative normalization.

### 3.3 GOALRANK

Based on the above insights, we propose a practical training framework for large ranking models in real-world recommendation scenarios.

**Reward modeling.** Following Zhang et al. (2025b), we first train a reward model $\hat{r}$ using real user feedback data, which can estimate the expected feedback for a given recommendation list (details are provided in Appendix B).

**Generator and policy.** As illustrated in Figure 2, given a generator $g_\theta$, the corresponding ranking policy is defined as $\pi_\theta := \mathrm{softmax} \circ g_\theta$. Conditioned on user context $\mathcal{X}_u$ and the candidate item set $\mathcal{V}_u$ provided by the preceding stage, the generator produces a recommendation list as

$$l_u^\theta = \arg\max_l \pi_\theta(l \mid \mathcal{X}_u, \mathcal{V}_u). \tag{6}$$

Note that this framework is model-agnostic: the generator can be instantiated by any sequence generation model.

---

[1] Here, the reward value refers to the user's actual feedback to a list, e.g., watch time or interaction behaviors.

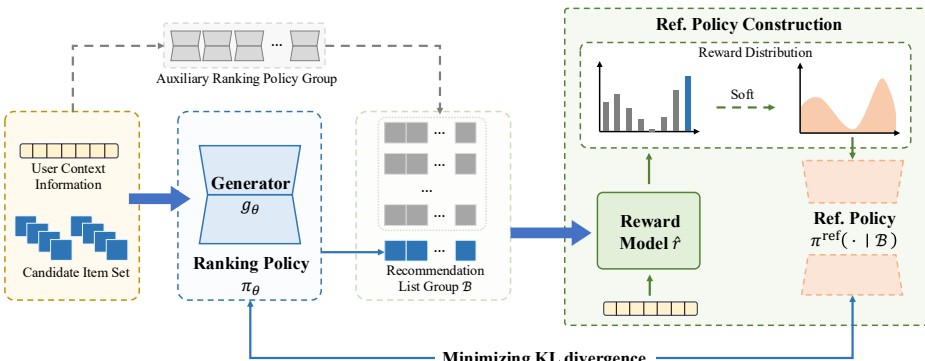

Figure 2: Training pipeline of group-relative optimization for a large ranker, GoalRank.

**Group construction.** As discussed in the previous section, constructing effective groups requires sufficiently large reward gaps among lists within each group, which is difficult to achieve when sampling multiple lists from a single generator. To address this, we introduce an auxiliary set of ranking policies $\mathcal{M}$ (including heuristic methods and lightweight neural models with implementation details provided in Appendix C). For each user $u$, we then construct a group of recommendation lists as

$$\mathcal{B}_u = \{l_u^\theta\} \cup \{l_u^i \mid l_u^i = \arg\max_l \pi_i(l \mid \mathcal{X}_u, \mathcal{V}_u), \ \pi_i \in \mathcal{M}\}.$$

As an additional option, all lists in $\mathcal{B}_u$ can be ranked by their rewards, and a uniformly sampled subset can then be selected, which further enforces larger reward gaps within the group and strengthens the validity of the condition in Equation 3.

**Training.** Given $\mathcal{B}_u$, we compute the reference policy $\pi^{\text{ref}}(\cdot \mid \mathcal{B}_u)$ via Equation 4 and optimize $g_\theta$ by minimizing the loss in Equation 5, instantiated with user-specific groups $\{\mathcal{B}_u\}_{u \in \mathcal{U}}$. This realizes the group-relative principle and provides a practical path to align $\pi_\theta$ with the oracle policy structure using accessible signals.

## 4  EXPERIMENT

In this section, we present both offline and online experiments to evaluate the effectiveness of GoalRank, which are designed to address the following research questions:

- **RQ1:** How does **GoalRank** perform on $N{\to}L$ ranking tasks compared with state-of-the-art baselines, and does it exhibit scaling behavior as model or data size increases?
- **RQ2:** How do (i) the size of recommendation list group $\mathcal{B}$ and (ii) the reward model's prediction bias affect **GoalRank** performance?
- **RQ3 (online):** How does **GoalRank** perform in real-world industrial recommendation scenarios?

### 4.1  OFFLINE EXPERIMENTS

#### 4.1.1  DATASETS AND OFFLINE EXPERIMENTS SETTING

We conduct offline experiments on two public datasets, ML-1M (Harper, 2015) and Amazon-Book (McAuley et al., 2015), as well as two datasets of different scales collected from our industrial short-video platform, denoted as Industry and Industry-0.1B. The statistics of the four preprocessed datasets are summarized in Appendix D.1.

For dataset construction, we first perform an 80/20 temporal split. For each user's interaction history (sorted chronologically), the task is framed as an $N{\to}L$ list-generation ranking problem with $N = 50$ and $L = 6$. Specifically, we use a pre-trained Matrix Factorization (MF) model (Koren et al., 2009) as the retriever to select the top-50 candidate items for each user. The last six interactions in each user's historical sequence are treated as ground truth, representing the target list after ranking. For industry datasets, we define a long view (watching a video for more than 85% of its duration) as a positive signal, indicating meaningful user–item engagement.

Table 1: Overall performance of different ranking methods. The highest scores are in bold, and the runner-ups are with underlines. All improvements are statistically significant with student t-test $p < 0.05$."Improv." denotes the improvements over the best baselines.

| | Methods | ML-1M | | | | | Industry | | | | | Book | | | | |
|---|---|---|---|---|---|---|---|---|---|---|---|---|---|---|---|---|
| | | H@6 | N@6 | M@6 | F1@6 | AUC | H@6 | N@6 | M@6 | F1@6 | AUC | H@6 | N@6 | M@6 | F1@6 | AUC |
| G-only | DNN | 56.86 | 70.30 | 59.28 | 62.16 | 86.87 | 37.32 | 54.56 | 42.38 | 43.72 | 74.73 | 60.28 | 69.61 | 58.58 | 62.45 | 83.02 |
| | DLCM | 62.31 | 73.87 | 63.82 | 67.96 | 89.35 | 39.69 | 60.67 | 48.90 | 46.61 | 75.80 | 66.80 | 75.88 | 65.39 | 69.28 | 91.93 |
| | PRS | 59.35 | 73.10 | 62.51 | 64.72 | 88.84 | 44.75 | 64.39 | 51.88 | 52.59 | 89.93 | 66.15 | 75.70 | 64.84 | 68.64 | 92.01 |
| | PRM | 60.09 | 72.85 | 62.21 | 65.51 | 88.20 | 39.92 | 55.93 | 42.97 | 46.18 | 85.15 | 67.86 | 76.88 | 66.44 | 70.42 | 92.00 |
| | MIR | 62.22 | 74.33 | 64.47 | 67.97 | 87.76 | 37.01 | 55.79 | 43.16 | 44.50 | 79.95 | 66.08 | 71.48 | 56.62 | 68.62 | 91.82 |
| | RankMixer | 60.88 | 72.65 | 62.68 | 64.18 | 92.47 | 49.72 | 69.19 | 58.73 | 60.24 | 91.03 | 68.03 | 76.45 | 66.27 | 71.26 | 92.23 |
| G-E | EGRank | 62.76 | 74.75 | 64.97 | 68.46 | 88.72 | 40.09 | 59.01 | 47.52 | 47.06 | 77.44 | 70.73 | 80.75 | 72.40 | 73.33 | 89.40 |
| | PIER | 62.74 | 75.99 | 65.98 | 68.74 | 90.43 | 45.35 | 65.11 | 52.55 | 53.35 | 90.93 | 71.14 | 80.22 | 71.62 | 73.74 | 92.26 |
| | NAR4Rec | 62.81 | 75.01 | 65.42 | 68.31 | 88.30 | 44.31 | 63.83 | 51.45 | 52.08 | 89.94 | 70.08 | 79.46 | 70.69 | 72.66 | 92.44 |
| MG-E | G-3 | 55.51 | 67.39 | 55.52 | 55.51 | 60.73 | 49.42 | 68.29 | 56.23 | 55.50 | 83.44 | 68.76 | 76.36 | 65.82 | 71.33 | 85.44 |
| | G-20 | 58.66 | 69.86 | 58.60 | 64.18 | 81.76 | 52.66 | 70.70 | 59.02 | 61.81 | 76.46 | 72.99 | 78.68 | 68.66 | 75.72 | 77.07 |
| | G-100 | 60.64 | 70.97 | 59.93 | 66.29 | 76.48 | 55.77 | 72.35 | 60.95 | 64.27 | 75.30 | 77.21 | 82.15 | 73.78 | 80.09 | 77.36 |
| | GoalRank | 73.56↑ | 83.43↑ | 76.16↑ | 80.15↑ | 97.64↑ | 69.93↑ | 86.93↑ | 79.01↑ | 82.29↑ | 98.07↑ | 80.35↑ | 84.88↑ | 77.91↑ | 83.44↑ | 94.46↑ |
| | Improv. | +17.12% | +9.79% | +15.43% | +16.60% | +5.59% | +25.39% | +20.15% | +29.63% | +28.04% | +7.73% | +4.07% | +3.32% | +5.60% | +4.18% | +2.19% |

Following common practices, we report Hit Ratio@L (H@L), NDCG@L (N@L), MAP@L (M@L), F1@L, and AUC with $L = 6$. Reported results are averaged over five independent runs.

### 4.1.2 BASELINES

We compare **GoalRank** against representative state-of-the-art methods from:

- **Generator-only methods**: These approaches rely on a single generator to produce item scores and directly generate the ranking list. Simple item-wise scoring models such as DNN (Covington et al., 2016) estimate user feedback independently for each user–item pair. More advanced methods, including DLCM (Ai et al., 2018), PRS (Feng et al., 2021a), PRM (Pei et al., 2019a), and MIR (Xi et al., 2022), explicitly capture mutual dependencies among candidate items. We additionally compare with RankMixer (Zhu et al., 2025).

- **Generator–Evaluator methods**: These methods (e.g., EGRerank (Huzhang et al., 2021), PIER (Shi et al., 2023), and NAR4Rec (Ren et al., 2024b)) first generate multiple reranked candidate lists and then leverage an evaluator to select the most effective one for the user. Following Yang et al. (2025), for PIER, we first apply a pointwise ranking model to select the top-6 items, enumerate all possible permutations, and then use the evaluator to identify the optimal ranking.

- **Multi-Generator–Evaluator methods**: These approaches ensemble multiple generators to expand the candidate-list space and enhance ranking performance (Yang et al., 2025). We evaluate this strategy under different ensemble sizes, with the number of generators set to 3, 20, and 100.

To ensure fairness, all baselines are tuned within their respective parameter spaces. Unless otherwise specified (e.g., in scaling law experiments), the hidden embedding dimension of all models is fixed at 128, and model depths are kept consistent.**Moreover, all baselines share exactly the same evaluator (reward model) as GoalRank.** Additional details on baseline configurations and the architecture of GoalRank are provided in Appendix D.2.

### 4.1.3 MAIN RESULTS (RQ1)

**Ranking Performance.** Table 1 reports the overall results across three datasets. We highlight:

- **GoalRank consistently achieves the best performance.** GoalRank outperforms all baselines. On ML-1M, it improves H@6 and M@6 by +17.12% and +15.43%; on the Industry dataset, the gains are even more pronounced, reaching +25.39% in H@6 and +29.63% in M@6. These results confirm that a single large generator can better capture ranking signals than multi-stage model.

- **Two-stage G-E methods outperform early G-only approaches.** Models such as PIER and NAR4Rec surpass DNN, DLCM, and PRM by explicitly modeling list-wise utility, validating the effectiveness of evaluators over greedy generation.

- **MG-E further improves but quickly saturates.** As generator count increases from 3 to 100, H@6 rises from 49.42 to 55.77 on Industry. However, the gains diminish rapidly, and even the

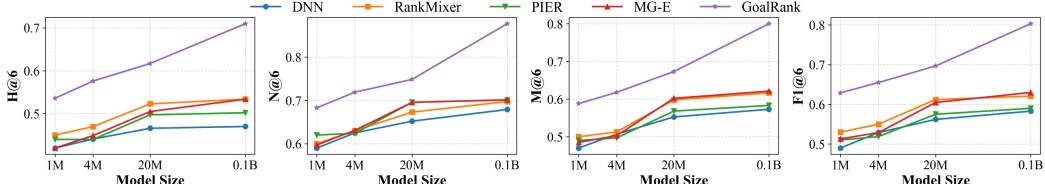

Figure 3: Scaling performance of GoalRank and baselines on the Industry-0.1B dataset across model sizes from 1M to 0.1B parameters.

Table 2: Performance with varying group sizes $|\mathcal{B}|$.

| $|\mathcal{B}|$ | 3 | 5 | 8 | 10 | 20 | 50 | 80 | 100 |
|---|---|---|---|---|---|---|---|---|
| H@6 | 62.88 | 64.52 | **69.95** | 69.94 | 69.93 | 67.34 | 63.29 | 63.50 |
| N@6 | 81.76 | 83.42 | 86.82 | **87.17** | 86.93 | 85.33 | 82.76 | 82.94 |
| M@6 | 72.50 | 74.44 | 78.86 | **79.34** | 79.01 | 76.96 | 73.69 | 73.92 |
| F1@6 | 74.00 | 75.94 | 82.25 | 82.28 | **82.29** | 79.26 | 74.47 | 74.76 |
| AUC | 97.50 | 97.79 | 98.05 | **98.15** | 98.07 | 98.06 | 97.75 | 97.77 |

Table 3: Performance with varying bias level $\lambda$.

| $\lambda$ | H@6 | N@6 | M@6 | F1@6 | AUC |
|---|---|---|---|---|---|
| 0.0 | 69.93 | 86.93 | 79.01 | 82.29 | 98.07 |
| 0.2 | 65.32 | 83.21 | 74.33 | 76.89 | 97.90 |
| 0.5 | 63.77 | 82.75 | 73.79 | 75.05 | 97.73 |

strongest MG-E models remain far below GoalRank, underscoring the inefficiency of enlarging the candidate set alone.

**Scaling Performance.** In Section 3.1, we showed theoretically that GoalRank admits scaling laws. Here, we empirically validate this by varying hidden dimensions, layer depth, and attention heads, adjusting model size from 1M to 0.1B. We compare GoalRank with four representative baselines: DNN, RanMixer, PIER, and MG-E. For fairness, baselines are scaled in the same manner as Goal-Rank, while the size of MG-E is increased by enlarging the number of generators.

Figure 3 presents the scaling performance on the Industry-0.1B dataset. We exclude AUC since GoalRank already achieves values above 0.98 even at small model sizes, though we observe further improvements as the model size increases, approaching 1.0. The model size ranges from 1M to 0.1B parameters.[2] We summarize the key findings as follows:

- **GoalRank demonstrates strong scaling.** Metrics improve steadily from 1M to 0.1B, with the sharpest gains between 10M and 0.1B, confirming clear scaling laws.
- **Baselines show weak scaling.** Larger sizes yield only modest improvements, reflecting the inherent limitation of pointwise scoring in approximating the optimal policy.
- **MG-E saturates.** Adding generators helps initially but plateaus quickly, indicating diminishing returns compared with GoalRank's single-model scaling.

### 4.1.4 ABLATION STUDY (RQ2)

In this section, we present ablation studies to examine two key factors: (i) the impact of the group size $|\mathcal{B}|$ on constructing the reference policy, and (ii) the robustness of GoalRank under varying levels of bias in the reward model. We report results on the Industry dataset, which is representative of the overall trends observed across all datasets.

**Influence of the Size of $\mathcal{B}$.** Table 2 reports the performance of GoalRank under different group sizes $|\mathcal{B}|$. We observe that very small groups (3–5) fail to provide enough samples for constructing a reliable reference policy, while overly large groups (50–100) weaken the reward gaps mentioned in Equation 3 and thus amplify the bias of the reward model. The best performance is achieved with moderate group sizes (8–20), which strike a balance between sample sufficiency and bias mitigation. Importantly, GoalRank consistently outperforms the best baseline even when $|\mathcal{B}|$ is set suboptimally.

**Influence of Prediction Bias of Reward Models.** As discussed in Section 3.2, the reward model $\hat{r}$ used to construct the reference policy is inevitably biased. To examine the effect of this bias, we introduce controlled noise by defining

$$\hat{r}_{\text{bias}=\lambda}(l) := (1-\lambda)\hat{r}(l) + \lambda\varepsilon, \qquad \varepsilon \sim \mathcal{N}(0,1),$$

---

[2]For very small models, training on the full dataset leads to unstable convergence. To ensure fair comparison, we proportionally sample the dataset for all models (including GoalRank) at the same parameter scale.

Table 4: Online performances improvement of GoalRank. All results are statistically significant.

| Method | APP Stay Time | Watch Time | Effective View | Like | Comment |
|---|---|---|---|---|---|
| **GoalRank + MG-E** *v.s.* **MG-E** | 0.092% | 0.111% | 0.836 % | 0.228% | 0.506% |
| **GoalRank** *v.s.* **MG-E** | 0.149% | 0.197% | 1.212% | 0.227% | 0.802% |

and evaluate GoalRank under different bias levels $\lambda \in \{0.0, 0.2, 0.5\}$. Results in Table 3 show that performance degrades only slightly as $\lambda$ increases, indicating that GoalRank is robust to reward model bias. Remarkably, even with $\lambda = 0.5$, GoalRank still outperforms state-of-the-art baselines.

## 4.2 LIVE EXPERIMENTS

### 4.2.1 EXPERIMENTAL SETUP

We evaluate GoalRank through a large-scale online A/B test on a real-world short-video recommendation platform. The platform serves over half a billion daily active users, with an item pool of tens of millions of videos. The system follows a two-stage workflow: (i) retrieval, which selects $N$ candidate items from millions, and (ii) ranking, which generates a final recommendation list of length $L$. GoalRank is deployed in the ranking stage, where $N = 120$ candidates are provided and $L = 6$ items are exposed to each user. The workflow and latency is illustrated in Figure 4 (Appendix).

### 4.2.2 EVALUATION PROTOCOL

We randomly partition online traffic into eight buckets, each covering about one-eighth of total users (tens of millions per bucket). We compare three settings: the production MG-E baseline(which consists of tens of generator models and hundreds of candidate lists), a hybrid setting where GoalRank serves 30% of the traffic alongside MG-E, and a pure GoalRank deployment. Each A/B test runs for at least 14 days to ensure statistical reliability. We track standard business metrics, including *App Stay Time* (a key overall engagement indicator), *Watch Time* (average continuous viewing length), *Effective Views* (total view count), and other behavior-specific rates on recommended items.

### 4.2.3 RESULTS ANALYSIS

We report the results of the two-week online A/B tests in Table 4. GoalRank consistently outperforms the production MG-E framework across all business-critical metrics. Even in the hybrid setting (GoalRank + MG-E), we observe significant gains, while a full deployment of GoalRank yields the largest improvements. These results demonstrate that GoalRank can not only complement but also fully replace the existing MG-E framework, providing superior ranking quality without incurring trade-offs. GoalRank + MG-E has been deployed to serve the full user traffic in production.

## 5 CONCLUSION

In this work, we revisit the design of ranking models in recommender systems and challenged the prevailing (Multi-)Generator–Evaluator paradigm. We theoretically proved that, for any (finite Multi-)Generator–Evaluator model, there always exists a generator-only ranker that achieves strictly smaller approximation error to the optimal ranking policy, and that this error decreases further as model size grows. Building on this result, we derived an evidence upper bound of the one-stage objective and introduced the group-relative optimization principle, which leverages a reward model trained on real user feedback to construct a reference policy and provides a practical training objective for generator-only rankers. We instantiated these insights in **GoalRank**, a large generator-only ranker optimized under the proposed principle. Extensive offline and online experiments demonstrated that **GoalRank** consistently outperforms SOTA methods and exhibits clear scaling behavior.

**Limitation and Future Work.** In real-world applications, ranking often needs to accommodate diverse and frequently changing business objectives. Compared with (Multi-)Generator–Evaluator models, a generator-only framework like GoalRank is less flexible in adapting to such shifts. A promising direction is to incorporate business-specific contextual signals into GoalRank, thereby enhancing its adaptability and generalization across objectives. Moreover, recent progress in large recommendation models has demonstrated remarkable success in the retrieval stage. However, most of these efforts overlook list-wise modeling, limiting their ability to capture the benefits unique to

ranking. Future work may therefore explore how large retrieval and large ranking models can be jointly optimized to build more powerful end-to-end recommender systems.

**Reproducibility Statement.** For the theoretical results, detailed derivations are provided in Appendix A. For the empirical studies, we will release the implementation and training code at `https://github.com/Kaike-Zhang/GoalRank` to ensure reproducibility.

**Ethics Statement.** This work aims to improve the ranking stage of recommender systems to enhance user satisfaction. It does not raise any specific ethical concerns.

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

# A PROOF OF THEOREM 1

**Notation.** Recall the space of the ranking list in Section 2. For simplicity and clarity, in this section, we omit the $u \in \mathcal{U}$ in the expressions of the following symbols. The space of ranking list $\mathcal{L}$ is:

$$\mathcal{L} := \left\{ l = (v_1, \ldots, v_L) \in \mathcal{V}^L \mid v_i \neq v_j \ (i \neq j) \right\}, \quad |\mathcal{L}| = \mathrm{P}(N, L) := \frac{N!}{(N-L)!}.$$

Then the state set during the generation process can be expressed as:

$$\mathcal{S} := \left\{ (i, l_{<i}) \mid i \in [L], \ l_{<i} \in \mathcal{V}^{i-1}, \ v_a \neq v_b \ (a \neq b) \right\}.$$

and corresponding actions at state $(i, l_{<i})$:

$$\mathcal{A}(i, l_{<i}) := \mathcal{V} \setminus \{v_1, \ldots, v_{i-1}\}, \quad |\mathcal{A}(i, l_{<i})| = N - (i-1).$$

Given a score vector $a \in \mathbb{R}^{\mathcal{V}}$ and a feasible set $\mathcal{A} \subseteq \mathcal{V}$, the *masked softmax* is

$$\left(\mathrm{softmax}_{\mathcal{A}}(a)\right)_j := \begin{cases} \dfrac{e^{a_j}}{\sum_{t \in \mathcal{A}} e^{a_t}}, & j \in \mathcal{A}, \\ 0, & j \notin \mathcal{A}. \end{cases}$$

For a finite index set $\mathcal{I}$ and $x \in \mathbb{R}^{\mathcal{I}}$, write $\|x\|_{\infty; \mathcal{I}} := \max_{i \in \mathcal{I}} |x_i|$. We use $\mathrm{KL}(\cdot \| \cdot)$ for Kullback–Leibler divergence.

**Lemma 1.** *Let $k \in \mathbb{N}$ and $g_{m,i} \in \mathcal{G}_m$ $(i = 1, \ldots, k)$ be $k$ generators. Assume each $g_{m,i}$ is locally Lipschitz in its parameter $\theta_i \in [A, B]^{d_m}$, and the masking set $\mathcal{A}(t, l_{<t})$ depends only on $(t, l_{<t})$ (not on parameters). For mixing weights $\boldsymbol{\omega} = [\omega_i]_{i=1}^k \in \Delta^{k-1}$, define*

$$\Phi : \ \Theta_k \to \Delta^{|\mathcal{L}|-1}, \qquad \Theta_k := \left([A, B]^{d_m}\right)^k \times \Delta^{k-1}, \tag{7}$$

*where*

$$\Phi(\boldsymbol{\theta}_{1:k}, \boldsymbol{\omega}) = \sum_{i=1}^k \omega_i \, \pi_{g_{m,i}},$$

*where $\pi_{g_{m,i}} = \left(\pi_{g_{m,i}}(l)\right)_{l \in \mathcal{L}}$ is the masked-softmax autoregressive policy induced by $g_{m,i}$. Then $\mathcal{C}_m^k := \mathrm{im}(\Phi)$ satisfies:*

1. *$\mathcal{C}_m^k$ is compact;*

2. *$\dim_{\mathrm{Haus}}\left(\mathcal{C}_m^k\right) \leq \min\left\{ k d_m + (k-1), \ |\mathcal{L}| - 1 \right\}$.*

*Proof.* **Compactness.** Each parameter domain $[A, B]^{d_m}$ is compact, and finite Cartesian products preserve compactness; the simplex $\Delta^{k-1}$ is also compact. **Hence $\Theta_k$ is compact.** For a fixed list $l \in \mathcal{L}$ and position $t$, given the smoothness of softmax and $g_{m,i}$ is $C^1$, the masked-softmax distribution

$$\pi_{g_{m,i}}(l) = \prod_{t=1}^L \frac{\exp\left(z_{t,l_t}^{(i)}(l_{<t})\right)}{\sum_{j \in \mathcal{A}(t, l_{<t})} \exp\left(z_{t,j}^{(i)}(l_{<t})\right)}, \qquad z_{t, \cdot}^{(i)} = g_{m,i}(t, l_{<t}, \cdot; \theta_i),$$

is $C^1$ in $\theta_i$, as masking only discards coordinates independent of parameters.

Since $\Phi$ is a convex combination of such policies, it is $C^1$ in $(\boldsymbol{\theta}_{1:k}, \boldsymbol{\omega})$, hence continuous. By continuity of $\Phi$ on the compact domain $\Theta_k$, its image $\mathcal{C}_m^k$ **is compact**.

**Dimension bound.** On $\Theta_k$ (compact), each map $(\theta_i, \boldsymbol{\omega}) \mapsto \omega_i \pi_{g_{m,i}}$ is Lipschitz: $\theta_i \mapsto z^{(i)}$ is Lipschitz; composition with masked-softmax is Lipschitz (bounded Jacobian on the relevant compact image); the product over $t$ and convex mixing in $\boldsymbol{\omega}$ preserve Lipschitzness since all factors are uniformly bounded. Thus $\Phi$ is Lipschitz on $\Theta_k$. Lipschitz maps do not increase Hausdorff dimension, yielding

$$\dim_{\mathrm{Haus}}(\mathrm{im}(\Phi)) \leq \dim_{\mathrm{Haus}}(\Theta_k) = k d_m + (k-1),$$

and trivially $\dim_{\mathrm{Haus}}(\mathrm{im}(\Phi)) \leq |\mathcal{L}| - 1$, giving the stated minimum. $\qquad \square$

**Theorem 2.** *Let $\mathcal{V}$ be the candidate set with $N := |\mathcal{V}|$, and let $\mathcal{L}$ be the set of length-$L$ lists without repetition, so $d := |\mathcal{L}| - 1 = \mathrm{P}(N, L) - 1$. Let $\mathcal{C}_m^k(\alpha, \beta) \subset \Delta^d$ be the $k$-mixture $(\alpha, \beta)$-bounded policy space. Assume Lemma 1 holds with*

$$\dim_{\mathrm{Haus}}\big(\mathcal{C}_m^k(\alpha, \beta)\big) \ \leq \ r := k\, d_m + (k - 1) \qquad and \qquad r < d.$$

*Then for Lebesgue-almost every fully supported target policy $\pi^* \in \mathrm{int}(\Delta^d)$,*

$$\inf_{\pi \in \mathcal{C}_m^k(\alpha, \beta)} \mathrm{KL}\big(\pi^* \| \pi\big) \ > \ 0. \tag{8}$$

*Proof.* By Lemma 1, $\mathcal{C}_m^k(\alpha, \beta)$ is compact and $\dim_{\mathrm{Haus}}(\mathcal{C}_m^k(\alpha, \beta)) \leq r < d = \dim(\mathrm{aff}(\Delta^d))$. Hence the $d$-dimensional relative Lebesgue measure (equivalently, the $d$-dimensional Hausdorff measure) of $\mathcal{C}_m^k(\alpha, \beta)$ inside $\mathrm{aff}(\Delta^d)$ is zero. Therefore, for Lebesgue-a.e. $\pi^* \in \mathrm{int}(\Delta^d)$ we have $\pi^* \notin \mathcal{C}_m^k(\alpha, \beta)$.

Fix such a $\pi^*$ with $\pi^* \succ 0$. Every $\pi \in \mathcal{C}_m^k(\alpha, \beta)$ assigns strictly positive probability to each $l \in \mathcal{L}$, so the map

$$\Psi(\pi) \ := \ \mathrm{KL}\big(\pi^* \| \pi\big) \ = \ \sum_{l \in \mathcal{L}} \pi^*(l) \, \log \frac{\pi^*(l)}{\pi(l)}$$

is finite and continuous on the compact set $\mathcal{C}_m^k(\alpha, \beta)$; thus the minimum

$$\delta \ := \ \min_{\pi \in \mathcal{C}_m^k(\alpha, \beta)} \mathrm{KL}\big(\pi^* \| \pi\big)$$

is attained. Since $\mathrm{KL}(\pi^* \| \pi) = 0$ iff $\pi = \pi^*$ and $\pi^* \notin \mathcal{C}_m^k(\alpha, \beta)$, we must have $\delta > 0$, proving Equation 8. $\qquad\square$

Classical universal approximation results (Cybenko, 1989; Hornik et al., 1989; Hornik, 1991; Leshno et al., 1993; Sonoda & Murata, 2017) yield the following lemma. For clarity and generality, we adopt MLP-based generators as the foundational model class. The goal is to show that even the most basic architecture—the MLP—already has sufficient expressive power to approximate the target policy. Universal approximation for Transformers (and related architectures) in sequence modeling is also known; see Yun et al. (2019); Augustine (2024).

**Lemma 2.** *Let $K \subset \mathbb{R}^n$ be compact and let $F : K \to \mathbb{R}^m$ be continuous. Let $\phi : \mathbb{R} \to \mathbb{R}$ be any continuous activation function that is not a polynomial on any interval.[3] Then for every $\eta > 0$ there exists a fixed-depth (e.g., one hidden layer), arbitrarily wide MLP $h_\Theta : K \to \mathbb{R}^m$ with activation $\phi$ such that*

$$\|h_\Theta - F\|_{\infty; K} \ < \ \eta.$$

*Consequently, for any finite Borel measure $\mu$ on $K$ and any $1 \leq p < \infty$, we also have $\|h_\Theta - F\|_{L^p(K, \mu)} < \eta$.*

**Remark 1** (Finite domains). *On a finite domain $K = \{x_1, \ldots, x_T\}$, continuity is automatic and*

$$\|h_\Theta - F\|_{\infty; K} = \max_{t \in [T]} \|h_\Theta(x_t) - F(x_t)\|_\infty,$$

*which coincides with the maximum pointwise error.*

To adapt Lemma 2 to the ranking setting, we encode the autoregressive state–action tuples into a compact Euclidean set.

**Corollary 1.** *Let*

$$\rho : \mathcal{D} \to \mathbb{R}, \qquad \mathcal{D} := \big\{(i, l_{<i}, j) \ : \ 1 \leq i \leq L, \ j \in \mathcal{A}(i, l_{<i})\big\},$$

*be defined by*

$$\rho(i, l_{<i}, j) \ := \ \log \pi^*(l_i = j \mid l_{<i}),$$

*where $\pi^*$ is a fully supported target autoregressive policy on the feasible sets $\mathcal{A}(i, l_{<i})$. Fix any injective encoding $\psi : \mathcal{D} \hookrightarrow K \subset \mathbb{R}^d$ with $K$ compact. Then for any $\sigma > 0$ there exists an MLP $h_\Theta : K \to \mathbb{R}$ such that*

$$\| h_\Theta \circ \psi - \rho \|_{\infty; \mathcal{D}} \ \leq \ \sigma.$$

---

[3]This covers common activations such as sigmoid, $\tanh$, ReLU, leaky-ReLU, ELU, and softplus.

*Proof.* The index set $\mathcal{D}$ is finite (since $N, L < \infty$), hence any function defined on $\psi(\mathcal{D}) \subset K$ is continuous w.r.t. the subspace topology. Apply Lemma 2 with $m = 1$ to obtain $h_\Theta$ with the desired uniform bound. $\qquad\square$

**Lemma 3.** *Let $p(j) \propto e^{\rho_j}$ on a finite set and $q(j) \propto e^{\rho_j + \varepsilon_j}$ with $|\varepsilon_j| \le \sigma$ for all $j$. Then for all $j$,*

$$e^{-2\sigma} \; \le \; \frac{q(j)}{p(j)} \; \le \; e^{2\sigma},$$

*and in particular*

$$\mathrm{KL}(p\|q) \; \le \; 2\sigma, \qquad \|p - q\|_1 \; \le \; e^{2\sigma} - 1.$$

*Moreover, for an autoregressive policy over length-$L$ lists,*

$$\mathrm{KL}\big(\pi^*\|\tilde{\pi}\big) \; = \; \sum_{i=1}^{L} \mathbb{E}_{l_{<i} \sim \pi^*}\Big[ \mathrm{KL}\big(p_i(\cdot \mid l_{<i})\|q_i(\cdot \mid l_{<i})\big)\Big] \; \le \; 2L\sigma.$$

**Fact 1.** *Fix a feasible set $\mathcal{A}$. Let $p = \mathrm{softmax}_{\mathcal{A}}(a)$ and $q = \mathrm{softmax}_{\mathcal{A}}(b)$ with $a, b \in \mathbb{R}^{\mathcal{V}}$. If $\|a - b\|_{\infty;\mathcal{A}} \le \sigma$, then*

$$\mathrm{KL}(p\|q) \; \le \; 2\sigma.$$

*Proof.* Write $\varepsilon_j := b_j - a_j$ for $j \in \mathcal{A}$, so $|\varepsilon_j| \le \sigma$. Then

$$\frac{q_j}{p_j} = \frac{e^{b_j} / \sum_{t \in \mathcal{A}} e^{b_t}}{e^{a_j} / \sum_{t \in \mathcal{A}} e^{a_t}} = \frac{e^{\varepsilon_j}}{\sum_{t \in \mathcal{A}} p_t e^{\varepsilon_t}} \in \left[\frac{e^{-\sigma}}{e^{\sigma}}, \frac{e^{\sigma}}{e^{-\sigma}}\right] = \left[e^{-2\sigma}, e^{2\sigma}\right].$$

Hence $\log \frac{p_j}{q_j} \le 2\sigma$ for all $j \in \mathcal{A}$, and

$$\mathrm{KL}(p\|q) = \sum_{j \in \mathcal{A}} p_j \log \frac{p_j}{q_j} \; \le \; \sum_{j \in \mathcal{A}} p_j \cdot 2\sigma \; = \; 2\sigma.$$

$\qquad\square$

**Fact 2.** *Let $\pi^*$ and $\pi$ be autoregressive list policies on $\mathcal{V}$ of length $L$ with conditionals supported on $\mathcal{A}(i, l_{<i})$. Then*

$$\mathrm{KL}(\pi^*\|\pi) = \sum_{i=1}^{L} \mathbb{E}_{l_{<i} \sim \pi^*}\Big[\mathrm{KL}\big(\pi^*(\cdot \mid l_{<i}) \,\|\, \pi(\cdot \mid l_{<i})\big)\Big].$$

*Proof.* By the chain rule,

$$\log \frac{\pi^*(l_1, \ldots, l_L)}{\pi(l_1, \ldots, l_L)} = \sum_{i=1}^{L} \log \frac{\pi^*(l_i \mid l_{<i})}{\pi(l_i \mid l_{<i})}.$$

Taking expectation w.r.t. $\pi^*$ yields

$$\mathrm{KL}(\pi^*\|\pi) = \sum_{i=1}^{L} \mathbb{E}_{l_{<i} \sim \pi^*} \, \mathbb{E}_{l_i \sim \pi^*(\cdot | l_{<i})}\Big[\log \frac{\pi^*(l_i \mid l_{<i})}{\pi(l_i \mid l_{<i})}\Big],$$

which is the stated form because the inner expectation equals $\mathrm{KL}\big(\pi^*(\cdot \mid l_{<i})\|\pi(\cdot \mid l_{<i})\big)$. $\qquad\square$

**Theorem 3.** *Let $\mathcal{G}_{\mathrm{MLP}}$ be the class of (fixed-depth, arbitrary-width) MLP generators that, given $(i, l_{<i})$, produce logits $g(i, l_{<i}, \cdot) \in \mathbb{R}^{\mathcal{V}}$. Define the induced policy class*

$$\mathcal{F}_{\mathrm{MLP}} := \big\{ \, \pi_\Theta \; : \; \pi_\Theta(\cdot \mid l_{<i}) = \mathrm{softmax}_{\mathcal{A}(i, l_{<i})}\big(g_\Theta(i, l_{<i}, \cdot)\big) \, \big\}.$$

*Then for every $\varepsilon > 0$ there exists a width $W(\varepsilon, N, L)$ and parameters $\Theta$ such that*

$$\mathrm{KL}(\pi^*\|\pi_\Theta) \; \le \; \varepsilon, \qquad \text{hence} \qquad \lim_{W \to \infty} \mathcal{E}(\mathcal{F}_{\mathrm{MLP}}) \; = \; 0,$$

*where $\mathcal{E}(\mathcal{F}) := \inf_{\pi \in \mathcal{F}} \mathrm{KL}(\pi^*\|\pi)$.*

*Proof.* Define target logits on the effective domain $\mathcal{D}$ by $\rho(i, l_{<i}, j) := \log \pi^*(l_i = j \mid l_{<i})$ for $j \in \mathcal{A}(i, l_{<i})$. Fix an injective encoding $\psi : \mathcal{D} \hookrightarrow K \subset \mathbb{R}^d$ into a compact $K$.

By Corollary 1, for any $\sigma > 0$ there exists an MLP $h_\Theta : K \to \mathbb{R}$ such that $\|h_\Theta \circ \psi - \rho\|_{\infty;\mathcal{D}} \le \sigma$. Use $g_\Theta(i, l_{<i}, j) := h_\Theta(\psi(i, l_{<i}, j))$ for $j \in \mathcal{V}$. Define the policy

$$\pi_\Theta(l_i = j \mid l_{<i}) := \mathrm{softmax}_{\mathcal{A}(i,l_{<i})}\big(g_\Theta(i, l_{<i}, \cdot)\big)_j.$$

For each prefix $l_{<i}$, Fact 1 with $a = \rho(i, l_{<i}, \cdot)$, $b = g_\Theta(i, l_{<i}, \cdot)$ yields

$$\mathrm{KL}(\pi^*(\cdot \mid l_{<i}) \,\|\, \pi_\Theta(\cdot \mid l_{<i})) \le 2\sigma.$$

Applying Fact 2 gives

$$\mathrm{KL}(\pi^* \| \pi_\Theta) = \sum_{i=1}^{L} \mathbb{E}_{l_{<i} \sim \pi^*}\big[\mathrm{KL}\big(\pi^*(\cdot \mid l_{<i}) \| \pi_\Theta(\cdot \mid l_{<i})\big)\big] \le 2L\sigma.$$

Choose $\sigma = \varepsilon/(2L)$ to obtain $\mathrm{KL}(\pi^* \| \pi_\Theta) \le \varepsilon$. Finally, by Lemma 2 (or the finiteness of $\mathcal{D}$), the achievable $\sigma$ tends to 0 as width $W \to \infty$, proving $\lim_{W \to \infty} \mathcal{E}(\mathcal{F}_{\mathrm{MLP}}) = 0$. $\qquad\square$

**Proposition 1.** *Let $\mathcal{G}_m(\alpha, \beta)$, $\mathcal{F}_m(\alpha, \beta)$, and $\mathcal{C}_m^k(\alpha, \beta)$ be as in Definitions 1–2. For $n \in \mathbb{N}_{>0}$, let the large-generator class be*

$$\mathcal{G}_M(\alpha, \beta, n) := \big\{ g_M \mid W(g_M) \ge k\alpha + n, \ D(g_M) \ge \beta \big\},$$

*with induced policy class $\mathcal{F}_M(\alpha, \beta, n) := \{\mathrm{softmax} \circ g_M : g_M \in \mathcal{G}_M(\alpha, \beta, n)\}$. Assume the list-generation domain has a finite effective index set $\mathcal{D} = \{(i, l_{<i}, j) : j \in \mathcal{A}(i, l_{<i})\}$ and the activation $\sigma$ enjoys a universal-approximation property on compact sets (e.g., standard MLP activations). Then*

$$\mathcal{C}_m^k(\alpha, \beta) \subseteq \overline{\mathcal{F}_M(\alpha, \beta, n)},$$

*where the closure is taken w.r.t. uniform convergence of masked conditionals on $\mathcal{D}$.*

*Proof.* Fix any mixture element $\pi_{\mathrm{mix}} \in \mathcal{C}_m^k(\alpha, \beta)$. By Definition 2, there exist generators $g_{m,r} \in \mathcal{G}_m(\alpha, \beta)$, $r = 1, \ldots, k$, with logits $z_r(i, l_{<i}, \cdot)$ on $\mathcal{A}(i, l_{<i})$ and mixture weights $\omega_r(i, l_{<i}) \ge 0$ with $\sum_{r=1}^{k} \omega_r(i, l_{<i}) = 1$ such that

$$\pi_{\mathrm{mix}}(\cdot \mid l_{<i}) = \sum_{r=1}^{k} \omega_r(i, l_{<i}) \, \mathrm{softmax}_{\mathcal{A}(i,l_{<i})}\big(z_r(i, l_{<i}, \cdot)\big).$$

Note that here we extend the mixture weights to be prefix-dependent, i.e., $\omega_r(i, l_{<i})$. This extension makes the proposition both stricter and more flexible, thereby broadening its generalization capability. If one wishes to exactly follow Definition 2, the weights can simply be degenerated to $\omega_r(l)$.

**Step 1 (block-diagonal embedding of the $k$ small generators).** Without loss of generality, pad each $g_{m,r}$ to width exactly $\alpha$ per hidden layer by adding zero weights/units. Construct a depth-$\beta$, width-$M$ network $g_M$ with $M \ge k\alpha + n$ whose hidden layers are partitioned into $k$ disjoint *generator blocks* of width $\alpha$ and one *evaluator block* of width $n$:

$$M = \underbrace{\alpha + \cdots + \alpha}_{k \text{ blocks}} + \underbrace{n}_{\text{evaluator}}.$$

For layers $1, \ldots, \beta - 1$, set the large-layer weights to be block-diagonal so that the $r$-th generator block exactly replicates the corresponding layer of $g_{m,r}$, and the evaluator block either copies its previous state or computes auxiliary features (details in Step 2). Thus, after $\beta - 1$ hidden layers, the first $k\alpha$ coordinates of the big network's hidden state equal the concatenation of the $\beta - 1$-th hidden activations of $\{g_{m,r}\}_{r=1}^{k}$.

**Step 2 (parameterizing the evaluator weights $\omega$ with $k-1$ degrees of freedom).** Use the evaluator block (of width $n \ge k - 1$) to produce *mixture logits* $u(i, l_{<i}) \in \mathbb{R}^k$ with the constraint that one coordinate is fixed as a reference (e.g., $u_k \equiv 0$), and define

$$\omega_r(i, l_{<i}) := \frac{e^{u_r(i, l_{<i})}}{\sum_{q=1}^{k} e^{u_q(i, l_{<i})}} \quad (r = 1, \ldots, k),$$

which realizes an arbitrary point in $\Delta^{k-1}$ through a $k-1$-dimensional parameterization. Because $\mathcal{D}$ is finite, the map $(i, l_{<i}) \mapsto \omega(i, l_{<i})$ can be approximated arbitrarily well by the evaluator block via UAT.

**Step 3 (combiner at depth $\beta$: realizing the log-sum-exp logits).** Define for $j \in \mathcal{A}(i, l_{<i})$ the target *combined* logits

$$\tilde{z}_j(i, l_{<i}) := \log \sum_{r=1}^{k} \frac{\omega_r(i, l_{<i})}{Z_r(i, l_{<i})} e^{z_{r,j}(i, l_{<i})}, \qquad Z_r(i, l_{<i}) := \sum_{t \in \mathcal{A}(i, l_{<i})} e^{z_{r,t}(i, l_{<i})}.$$

At the last hidden layer (the $\beta$-th nonlinear layer), allow *cross-block* connections from all generator blocks and the evaluator block into a width-$M$ hidden layer that serves as a single-hidden-layer approximator for the multivariate continuous mapping

$$\Phi : (z_1, \ldots, z_k, \omega) \mapsto \tilde{z} \quad \text{on the finite domain } \mathcal{D}.$$

By universal approximation, there exist weights in this last hidden layer (and the final linear readout) so that the resulting $g_M$ satisfies

$$\| g_M - \tilde{z} \|_{\infty; \mathcal{D}} \leq \sigma$$

for any prescribed $\sigma > 0$. Note that the depth requirement $D(g_M) \geq \beta$ is met (we used exactly $\beta$ nonlinear layers), and the width requirement $W(g_M) \geq k\alpha + n$ is used to house the $k$ embedded blocks ($k\alpha$ units) and the evaluator block ($n$ units).

**Step 4 (from logits to conditionals).** By the identity

$$\text{softmax}_{\mathcal{A}(i,l_{<i})}\big(\tilde{z}(i, l_{<i}, \cdot)\big) = \sum_{r=1}^{k} \omega_r(i, l_{<i}) \, \text{softmax}_{\mathcal{A}(i,l_{<i})}\big(z_r(i, l_{<i}, \cdot)\big),$$

the conditional produced by $\text{softmax}_{\mathcal{A}}(\tilde{z})$ equals the target mixture conditional. Since $\| g_M - \tilde{z} \|_{\infty; \mathcal{D}} \leq \sigma$ and the masked softmax is continuous, $\text{softmax}_{\mathcal{A}}(g_M)$ converges uniformly on $\mathcal{D}$ to $\text{softmax}_{\mathcal{A}}(\tilde{z}) = \pi_{\text{mix}}$ as $\sigma \downarrow 0$. Therefore $\pi_{\text{mix}} \in \overline{\mathcal{F}_M(\alpha, \beta, n)}$. Because $\pi_{\text{mix}}$ was arbitrary, the claimed inclusion holds. $\qquad \square$

**Remark 2** (Why $k\alpha + n$ and $k - 1$ neurons for $\omega$). *The $k\alpha$ term guarantees disjoint capacity to exactly embed the $k$ small generators via block-diagonal copying across the first $\beta-1$ hidden layers. The additional $n$ units form an evaluator head; choosing $n \geq k-1$ suffices to parameterize the simplex $\Delta^{k-1}$ via softmax logits $u \in \mathbb{R}^k$ with one fixed reference coordinate, while also providing enough width for the last-layer universal approximation of the log-sum-exp combiner.*

*Proof of Theorem 1.* **Step 1 (Coverage of $k$-mixtures by a single large generator).** By Proposition 1,

$$\mathcal{C}_m^k(\alpha, \beta) \subseteq \overline{\mathcal{F}_M(\alpha, \beta, n)},$$

where the closure is taken w.r.t. uniform convergence of masked conditionals on the finite effective domain. Because $\pi^*(l) > 0$ for all $l \in \mathcal{L}$ and $\mathcal{L}$ is finite, the map $\pi \mapsto \text{KL}(\pi^* \| \pi)$ is continuous under uniform convergence of the conditionals. Hence, for every $n$,

$$\mathcal{E}\big(\mathcal{F}_M(\alpha, \beta, n)\big) \leq \mathcal{E}\big(\mathcal{C}_m^k(\alpha, \beta)\big). \tag{9}$$

**Step 2 (Arbitrary accuracy by increasing width).** By Theorem 3 (UAT-backed policy approximation), for every $\varepsilon > 0$ there exists a fixed depth $L_0$ and a width threshold $W(\varepsilon, N, L)$, together with parameters $\Theta$, such that the induced policy $\pi_\Theta$ satisfies $\text{KL}(\pi^* \| \pi_\Theta) \leq \varepsilon$. Choose the fixed depth in Theorem 3 so that $L_0 \geq \beta$, which is allowed by the theorem. Then, taking $n$ large enough to ensure $k\alpha + n \geq W(\varepsilon, N, L)$, we have $\pi_\Theta \in \mathcal{F}_M(\alpha, \beta, n)$ and therefore

$$\mathcal{E}\big(\mathcal{F}_M(\alpha, \beta, n)\big) \leq \varepsilon. \tag{10}$$

Since $\varepsilon > 0$ was arbitrary, it follows that $\lim_{n \to \infty} \mathcal{E}(\mathcal{F}_M(\alpha, \beta, n)) = 0$.

**Step 3 (Strict improvement over the $k$-mixture space).** By Theorem 2, there exists $\delta > 0$ such that $\mathcal{E}(\mathcal{C}_m^k(\alpha, \beta)) = \delta$. Apply Step 2 with $\varepsilon := \delta/2$. Then for some $n_0$,

$$\mathcal{E}\big(\mathcal{F}_M(\alpha, \beta, n_0)\big) \leq \delta/2 < \delta = \mathcal{E}\big(\mathcal{C}_m^k(\alpha, \beta)\big).$$

By monotonicity in $n$ (the class $\mathcal{F}_M(\alpha, \beta, n)$ enlarges with $n$), the strict inequality holds for all $n \geq n_0$. Combining with equation 9 concludes the proof of both statements. $\qquad \square$

## B    REWARD MODEL TRAINING

Following Zhang et al. (2025b), we train a reward model to approximate user feedback on exposed recommendation lists. Let $u \in \mathcal{U}$ denote a user with context information $\mathcal{X}_u$ (e.g., historical interactions or side features). Suppose $l_u = (v_1, \ldots, v_{|l|})$ is the exposure list presented to user $u$, where $v_i \in \mathcal{V}_u \subseteq \mathcal{V}$ and $|l|$ is the list length. The corresponding real user feedback, such as watch time, clicks, or other engagement signals, is denoted by $r_{l_u} \in \mathbb{R}$.

The reward model is defined as a function

$$\hat{r} : \mathcal{X} \times \mathcal{V}^{|l|} \to \mathbb{R},$$

which, given the user context $\mathcal{X}_u$ and a candidate list $l_u$, predicts the expected feedback $\hat{r}(l_u \mid \mathcal{X}_u)$.

To train the reward model, we minimize the mean squared error between the predicted feedback $\hat{r}(l_u \mid \mathcal{X}_u)$ and the observed feedback $r_{l_u}$ across all users:

$$\mathcal{L}(\hat{r}) = \mathbb{E}_{u \in \mathcal{U}} \left[ \left( \hat{r}(l_u \mid \mathcal{X}_u) - r_{l_u} \right)^2 \right].$$

## C    METHODS FOR GROUP CONSTRUCTION

We consider multiple strategies for constructing groups of candidate lists, beyond the standard autoregressive sampling approach. The main methods are summarized as follows:

- **Autoregressive list generation (Jayaram & Thickstun, 2021):** The conventional approach samples a single list by generating the entire trajectory in an autoregressive generator. While effective in capturing dependencies, this method is relatively slow and produces only one list per sampling trajectory.

- **Parallel tree-structured generation (Jayaram & Thickstun, 2021; Wang et al., 2025a):** To improve efficiency and diversity, we allow the autoregressive generator to branch out in the first $K$ steps, forming a tree of partial sequences. The remaining $L - K$ steps are then completed deterministically, enabling parallel exploration of multiple candidate lists.

- **Softmax-based stochastic sampling (Holtzman et al., 2019; Efraimidis & Spirakis, 2006):** After the first step of scoring by generator, items are sampled probabilistically according to the softmax distribution of their scores, instead of deterministically selecting the top item, which encourages more diverse list generation.

- **Markov process approximation (Metropolis et al., 1953):** The list generation process is modeled as a Markov chain, where each step only conditions on the immediately preceding item rather than the full history. We exhaustively explore and score all possible two-item pairs in the first two steps and the remaining items in the list are sampled sequentially in a chain-like manner.

- **Random selection:** As a simple baseline, we randomly sample six items to form a candidate list without using model guidance.

- **Heuristic substitution (Wang et al., 2025b):** Starting from already sampled lists, we heuristically replace up to two items to create new candidate lists while maintaining partial consistency with existing ones.

- **Diversity-oriented generation (Yang et al., 2025) :** we generate lists that are explicitly encouraged to differ significantly from previously sampled lists, thereby enhancing the diversity of the candidate set.

## D    DETAILS OF OFFLINE EXPERIMENTS SETTINGS

### D.1    DETAILS OF DATASET

The ML-1M dataset is a widely used public benchmark in recommender systems, containing approximately 1 million ratings provided by over 6,000 users on more than 3,900 movies. The Amazon Books dataset is a large-scale collection of product reviews focused on books available on Amazon, consisting of about 2 million reviews from over 35,000 users spanning more than 39,000 books. The

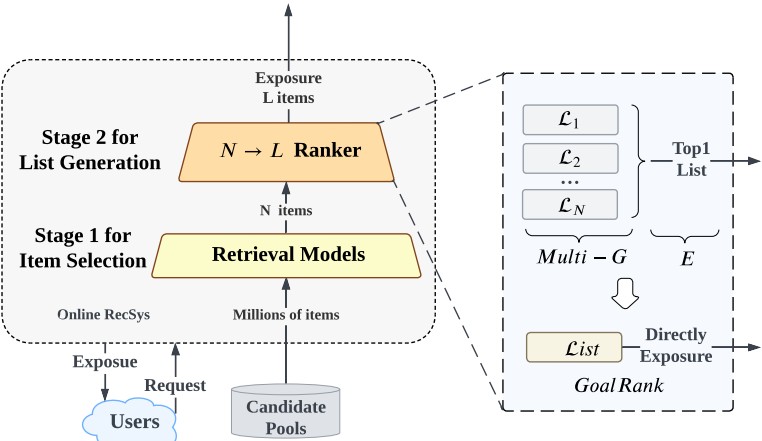

Figure 4: Online workflows.

Industry dataset is collected from a real-world short-video platform that serves over half a billion daily active users and hosts an item pool of tens of millions of videos. We construct two versions of this dataset: a smaller one with over 3 million interactions from 89,310 users on 10,395 videos, and a larger one with over 0.1 billion interactions from 1 million users on 0.3 million videos.

For preprocessing, all interactions are organized in chronological order, and users/items with fewer than 20 interactions are filtered out following the standard 20-core protocol Zhang et al. (2025a). We employ a Matrix Factorization (MF) model as the retriever simulator to generate candidate items for the ranking stage. The data is split into training and testing sets with a ratio of 8:2. For the ranking stage, interactions are sorted chronologically, and the last six interactions are used as the item list exposed to users after reranking. Table 5 reports the statistics of the processed datasets, including the number of users, items, interactions, and revealed lists.

Table 5: Dataset Statistics

| Dataset | $|\mathcal{U}|$ | $|\mathcal{I}|$ | # Interaction | # List |
|---|---|---|---|---|
| ML-1M | 6,020 | 3,043 | 995,154 | 161,646 |
| Amazon-Book | 35,732 | 38,121 | 1,960,674 | 311,386 |
| Industry | 89,310 | 10,395 | 3,270,132 | 513,010 |
| Industry-0.1B | 1,146,032 | 312,573 | 100,269,812 | 16,099,612 |

## D.2 DETAILS OF BASELINES

We detail the compared baselines of our main experiments in the following, covering three major categories: Generator-only methods, Generator–Evaluator methods, and Multi-Generator–Evaluator methods:

**Generator-only methods:** These approaches directly predict the item-wise scores for candidates and rank them accordingly, without explicit evaluation of whole lists.

- DNN (Covington et al., 2016) learn the user feedback for each user-item interaction.
- DLCM (Ai et al., 2018) refines initial rankings by leveraging local context from top-retrieved documents , using a recurrent neural network to capture document interactions and an attention-based loss function to capture item interactions.
- PRS (Feng et al., 2021a) also known as SetRank, which is a neural learning-to-rank model employs permutation-invariant neural ranking with multi-head self-attention to model cross-item dependencies, achieving robust performance across variable-length input sets.
- PRM (Pei et al., 2019a) addresses personalized re-ranking by integrating user-specific preferences into the re-ranking process, thus enhancing both personalization and relevance.
- MIR (Xi et al., 2022) captures complex hierarchical interactions between user actions and candidate list features to improve the accuracy of list-wise recommendation.

**Generator-Evaluator Methods:** These methods adopt a two-stage paradigm where the generator produces candidate lists and the evaluator selects the most promising one.

- EGRerank (Huzhang et al., 2021) proposes an evaluator–generator framework for e-commerce ranking. The evaluator estimates list utility given context, while the generator leverages reinforcement learning to maximize evaluator scores, with an additional discriminator ensuring evaluator generalization.
- PIER (Shi et al., 2023) follows a two-stage architecture consisting of a Fine-grained Permutation Selection Module (FPSM) and an Omnidirectional Context-aware Prediction Module (OCPM). The FPSM leverages SimHash to identify the top-$K$ candidate permutations based on user interests, while the OCPM evaluates these permutations through an omnidirectional attention mechanism.
- NAR4Rec (Ren et al., 2024b) introduces a non-autoregressive generative re-ranking model that alleviates data sparsity and candidate variability via contrastive decoding and unlikelihood training, while also considering its integration into broader generator–evaluator frameworks.

**Multi Generator-Evaluator:** These methods extend the generator–evaluator paradigm by ensembling multiple generators to enlarge the candidate list space and improve final ranking quality. For example, MG-E (Yang et al., 2025) aggregates outputs from multiple generators, each specializing in different candidate distributions, before applying evaluation for list selection.

To ensure a fair comparison, we adopt a relatively lightweight generator architecture for GoalRank. Specifically, the GoalRank Generator consists of two blocks: a lower block that performs feature crossing over all candidate items using several Transformer layers, and an upper block implemented as a Transformer decoder. During list generation, the decoder autoregressively predicts next-item scores over the full candidate set at each step, and GoalRank constructs the final list by sequentially selecting items (via Top-1 or sampling-based strategies).

### D.3 ONLINE LATENCY AND MFU

To demonstrate the efficiency and resource-utilization advantages of GoalRank's single-stage architecture, we report both the online latency and MFU of GoalRank compared with the existing Multi-Generator–Evaluator (MGE) pipeline. Notably, during training, many components of GoalRank can be executed in parallel. For example, the construction of auxiliary ranking-policy groups for reference-policy generation can be fully parallelized, resulting in negligible additional overhead. Under this setting:

- **Latency.** GoalRank achieves an online latency of **18.611 ms**, which is substantially faster than the multi-stage MGE pipeline (**34.235 ms**). This improvement stems from GoalRank's ability to directly generate the final list in a single stage, eliminating evaluator scoring and multiple candidate-list constructions .
- **MFU.** GoalRank attains an MFU of **12.65%**, compared with **2.03%** for the traditional two-stage MGE pipeline. The higher MFU reflects significantly better hardware utilization, ensuring that GoalRank does not increase overall training cost despite offering stronger performance.

These results validate that GoalRank is **practical, efficient, and deployment-ready**. Furthermore, **GoalRank has been successfully deployed in our online environment to serve full user traffic.**

### E USE OF LLMS

Large language models (LLMs) were employed to polish the main body of this paper. Their use was limited to grammar checking and correction of typographical errors.

