# OpenReview forum: "GoalRank: Group-Relative Optimization for a Large Ranking Model"
_ICLR.cc/2026/Conference — ICLR 2026 Poster_

### Official Review · Reviewer_LY4d · 2025-10-25

**Soundness:** 2
**Presentation:** 3
**Contribution:** 2
**Rating:** 4
**Confidence:** 4

**Summary:**

The paper proposes GoalRank, a novel framework for training a single-stage, large-scale ranking model to replace the inefficient traditional two-stage Generator-Evaluator (G-E) paradigm. The G-E paradigm suffers from performance saturation due to the vast combinatorial ranking space. GoalRank introduces the Group-Relative Optimization Principle, using a pre-trained Reward Model and relative normalization within a group of lists to construct a Reference Policy. This allows the single-stage model to be trained effectively, achieving a superior approximation of the optimal policy. The framework is theoretically proven to surpass finite G-E systems and validated on a massive industrial platform, showing significant lifts in core business metrics.

**Strengths:**

1. This work provides a strong theoretical proof that a sufficiently large single-stage ranker can achieve a strictly lower approximation error than any finite two-stage system, confirming its strong potential for scaling laws.

2. The Group-Relative Optimization is an practical solution for training against the unobservable optimal policy in large-scale production environments.

3. It achieved significant and convincing lifts in key business metrics during large-scale online A/B testing on a platform with over 500 million daily active users.

**Weaknesses:**

1. The paper’s academic context is weakened by placing related work in the appendix, and its scope is blurred between the ranking and re-ranking domains, leading to an incomplete set of comparative baselines. See details in Questions 2

2. The theoretical proof is contingent on the optimal strategy $\pi^{*}$, which is substituted by an approximate Reward Model $\hat{r}$ in practice. While the paper claims robustness against bias in $\hat{r}$, the inherent difficulty in training a truly optimal reward model means this approximation gap remains a potential bottleneck, limiting the performance ceiling achievable in the real world.

3. The overall reproducibility is a concern and the scalability experiments fails to demonstrate that other, more sophisticated single-stage solutions (beyond simple DNNs) cannot also achieve similar or superior scaling behavior.

**Questions:**

1. Placing the Related Work section entirely in the appendix detracts from the integrity of the main paper structure, making it difficult for the reader to immediately contextualize the work against existing literature during the initial review.

2. The paper's claimed field focus is sometimes ambiguous. While positioned as a solution for ranking, many of its baselines and related works cited (e.g., DLCM, PRM) are strongly associated with the re-ranking. If the work truly aims to target the ranking field, it should include comparisons against more foundational CTR-prediction methods and other contemporary Large Ranking Model (LRM) approaches like RankMixer: Scaling Up Ranking Models in Industrial Recommenders. If this work focuses on reranking, why not use some datasets and experimental settings more commonly used in the reranking field, such as Avito and Kuaishou?

3. Key engineering details, such as the specific construction of the Auxiliary Ranking Policy Group ($\mathcal{B}$) and the exact large model architecture, are either simplified or omitted.

4. The scalability experiments primarily focus on demonstrating superiority over the two-stage (G-E) paradigm, which quickly saturates. However, the comparison within the single-stage paradigm only includes simple DNN baselines. The paper fails to demonstrate that other, more sophisticated single-stage solutions (beyond simple DNNs) cannot also achieve similar or superior scaling behavior.

---

> ### Author Response · Authors · 2025-11-21
> **Rebuttal by Authors**
>
> Thank you for your detailed feedback and for acknowledging our **strong theoretical foundations**, **practical solution**,  **empirical validation on a real-world platform**, and  **extensive experiments**. We address your concerns below and sincerely hope these clarifications help re-evaluate our contributions.

---

> > ### Author Response · Authors · 2025-11-21
> > **Detailed Explanations (1/2)**
> >
> > ## The paper’s academic context is weakened by placing related work in the appendix
> >
> > Thank you for the suggestion. Due to strict page limits, we had to move the detailed related work analysis to the Appendix. In the **revised version (revised manuscript L103-L134),** we reorganize the layout to include the core related work discussion in the main text to strengthen the paper’s academic positioning. We appreciate your constructive feedback.
> >
> > * * *
> >
> > ## Difficulty of training an optimal reward model may limit performance in practice
> >
> > Thank you for this insightful question. We emphasize that  **GoalRank is explicitly designed to mitigate reward model biases**, supported by both  **theoretical justification**  and  **empirical evidence**.
> >
> > - **Theoretical Perspective**
> >   - As noted in the original version L230–L232 (revised manuscript L239-L241), we explicitly assume a potentially biased reward model $\hat{r}(l) = r^*(l) + b(l)$, where $b(l)$ denotes arbitrary noise or bias.
> >   - In the original version L233–L249 (revised manuscript L242-L249), we propose a  **group-relative ranking mechanism**, which converts absolute reward scores into  **relative partial orders**  within comparable groups. This procedure significantly reduces the effect of $b(l)$ by canceling out shared bias components within each group.
> >
> > - **Empirical Perspective**
> >   - In Table 3, we provide extensive experiments under  **severely biased reward models**.
> >   - We introduce bias via $\hat{r}\_{\text{bias}=\lambda} = (1 - \lambda)\hat{r}(l) + \lambda \epsilon,\quad \epsilon \sim \mathcal{N}(0,1)$, with $\lambda$= 0, 0.2, 0.5, introducing up to  **50% random noise**.
> >   - Even under large perturbations, GoalRank  **significantly outperforms all SOTA baselines**, demonstrating its robustness to reward bias in practice.
> >
> > * * *
> >
> > ## Concerns regarding reproducibility and the scalability experiment design
> > Thank you for raising these concerns.
> > - **Reproducibility.** We would like to clarify that **our full implementation is publicly available** at the original version L484–L485 (revised manuscript L483-L484) ([anonymousGitHub](https://anonymous.4open.science/R/GoalRank)) to ensure complete reproducibility of experiments.
> > - **Scalability experiments.** To address this concern, we additionally include **RankMixer**, a recent large-scale single-stage ranking model. Although RankMixer outperforms other one-stage baselines, it is not designed for list-wise generation. The updated scalability experiments in **Figure 3 of the revised manuscript** further confirm **GoalRank’s stronger and more stable scaling behavior**.

---

> > > ### Author Response · Authors · 2025-11-21
> > > **Detailed Explanations (2/2)**
> > >
> > > ## Ambiguity in field positioning: ranking vs. re-ranking, and baseline selection
> > >
> > > Thank you for pointing out this ambiguity. We clarify the positioning as follows.
> > >
> > > - Traditional **ranking** typically performs *point-wise* or *pair-wise* scoring over a large candidate pool of size $N$.
> > > * **Re-ranking** instead operates on a much smaller set of size $L$, producing an $L \to L$ reordered list.
> > > * In contrast, **GoalRank tackles the full end-to-end $N \to L$ list-generation problem**, i.e., generating an ordered list of length $L$ directly from a large candidate pool $N \gg L$.
> > >   In this sense, **GoalRank is an end-to-end ranking model** that *unifies* both:
> > >   * the **large-candidate scoring** stage of traditional ranking (operating over $N$), and
> > >   * the **list-wise ordering** stage of re-ranking (ordering $L$ selected items).
> > >
> > > Thus, the task formulation naturally **covers the output objectives of re-ranking** (list-wise ordering), while its **input scale and complexity align with full-ranking scenarios**.
> > >
> > > Because GoalRank’s outputs align with list-wise generation, re-ranking baselines are **more directly comparable** in terms of objective formulation. This is why our primary baselines focus on **list-wise and re-ranking models**, which share the same type of output space and evaluation protocol (i.e., list generation).
> > >
> > > To further strengthen the comparison, we additionally include **RankMixer**, a recent large-scale single-stage ranking model. Although RankMixer outperforms other one-stage baselines, it is not designed for list-wise generation. On the industry dataset (partial results shown below), GoalRank achieves **substantial improvements** across almost all metrics, and MGE-100 surpasses RankMixer on several metrics:
> > >
> > > | Method       | H@6       | N@6       | M@6       | F1@6      | AUC       |
> > > | ------------ | --------- | --------- | --------- | --------- | --------- |
> > > | RankMixer    | 49.72     | 69.19     | 58.73     | 60.24     | 91.03     |
> > > | MGE-3        | 49.42     | 68.29     | 56.23     | 55.50     | 83.44     |
> > > | MGE-100      | 55.77     | 72.35     | 60.95     | 64.27     | 75.30     |
> > > | **GoalRank** | **69.93** | **86.93** | **79.01** | **82.29** | **98.07** |
> > >
> > > The full set of results across all datasets is provided in Table 1 of the revised manuscript, where **GoalRank continues to outperform RankMixer and all baselines consistently**.
> > > * * *
> > >
> > > ## Missing engineering details: Auxiliary Ranking Policy Group and large-model architecture
> > >
> > > Thank you for pointing this out.
> > >
> > > - **Auxiliary Ranking Policy Group.** We clarify that the detailed construction of the **Auxiliary Ranking Policy Group** is provided in **Appendix D**. It includes several shallow ranking models as well as multiple heuristic list-generation strategies. Notably, the specific choice of auxiliary policies **does not materially affect GoalRank’s performance**: GoalRank only requires a recommendation-list group with sufficiently large reward gaps to construct reliable group-relative comparisons. We will explicitly reference this appendix section in the revised version to improve clarity.
> > >
> > > - **Large-model architecture.** The GoalRank Generator uses a lightweight two-block architecture. The lower block performs feature crossing over all candidate items using **multiple transformer layers**, while the upper block adopts a **transformer decoder** structure. During generation, the decoder autoregressively predicts the next-item scores from the full candidate set at each step, and GoalRank produces the final list by selecting (Top-1 or sampling-based) items sequentially.
> > >
> > > As noted in the original version L265–L266 (revised manuscript L270-L272), GoalRank is model-agnostic, and the central contribution of our work lies in the training framework, rather than proposing a new generator architecture. For this reason, we kept architectural details concise in the main method section, but we will clarify them more explicitly in the experimental section to improve readability.
> > >
> > >
> > > * * *
> > >
> > > Thank you again for your thorough and constructive review. We have responded to each concern and clarified misunderstandings. Based on these explanations, we respectfully request your reconsideration of the overall assessment. We believe that  **GoalRank advances the state of the art in large-scale ranking**  and offers both theoretical value and demonstrated real-world impact.

---

> > > > ### Author Response · Authors · 2025-11-25
> > > > **Kindly Reminder**
> > > >
> > > > Thank you for taking the time and effort to review our paper. We have carefully considered your comments and responded to each point with detailed clarifications and supporting evidence. As the deadline for the discussion period between authors and reviewers is approaching, we kindly ask whether our responses have addressed your concerns. We look forward to your feedback.
> > > >
> > > > Thank you once again for your time and effort.
> > > >
> > > > Sincerely,
> > > > The Authors

---

> > > > > ### Author Response · Authors · 2025-11-28
> > > > > **Gentle Reminder**
> > > > >
> > > > > Dear Reviewer LY4d,
> > > > >
> > > > > Thank you again for your detailed feedback and for acknowledging the strong theoretical foundations and practical value of our work. Your suggestions regarding the paper's structure and positioning have been incredibly helpful. **As the discussion period is ending soon**, we would like to ensure you have seen our response, specifically where we **reorganized the manuscript** and **added new experiments** to address your concerns.
> > > > >
> > > > > In our detailed response, we have:
> > > > >
> > > > > * **Revised Paper Structure:** As suggested, we moved the **Related Work section back to the main text** to strengthen the academic context.
> > > > > * **Clarified Robustness to Bias:** We explained how our **group-relative ranking** theoretically cancels out reward bias. We also highlighted experiments showing GoalRank maintains SOTA performance even under **50% reward noise**, addressing concerns about optimal reward models.
> > > > > * **Strengthened Baselines:** We added a comparison with **RankMixer** to address positioning and scalability concerns. GoalRank consistently outperforms it (e.g., **69.93 vs. 49.72 H@6**).
> > > > > * **Clarified Positioning & Details:** We clarified the **$N \to L$** end-to-end ranking formulation, provided architectural details of the Generator, and pointed to our **publicly available code** for reproducibility.
> > > > >
> > > > > We hope these revisions and clarifications resolve your concerns. We would greatly appreciate it if you could let us know if there are any remaining questions.
> > > > >
> > > > > Best regards,
> > > > > The Authors

---

### Official Review · Reviewer_JXWv · 2025-10-26

**Soundness:** 4
**Presentation:** 3
**Contribution:** 4
**Rating:** 8
**Confidence:** 3

**Summary:**

This paper proposes GoalRank, a novel ranking framework that replaces the complex Multi-Generator-Evaluator paradigm with a single large generator model. The contributions are threefold: theoretically, it proves that a sufficiently large generator can better approximate the optimal ranking policy than any finite M-G-E system, with error decreasing via scaling laws; methodologically, it introduces group-relative optimization, which trains the generator using robust relative comparisons within list groups rather than absolute rewards; empirically, large-scale offline and online experiments demonstrate consistent and significant improvements over state-of-the-art baselines, validating the framework's effectiveness and scalability.

**Strengths:**

1.  The paper provides a rigorous theoretical proof that a single large generator can surpass any finite M-G-E system, with a clear scaling law. This theory is strongly supported by empirical results, creating a cohesive narrative.
2. The proposed group-relative optimization is an innovative solution for training large rankers. It effectively leverages relative comparisons within groups to create a robust training signal that is resilient to reward model bias.
3. The paper offers thorough validation, from offline benchmarks to large-scale online A/B tests on a major platform. The consistent and significant improvements across both academic and business metrics convincingly demonstrate real-world impact.

**Weaknesses:**

1.  The training process itself is complex and resource-heavy, The proposed method shifts complexity to the training stage, requiring a costly and intricate pipeline, especially for large-scale industry recommendation systems.
2. The architectural details and specific implementation of the generator model itself are not sufficiently elaborated, making it difficult to fully assess the proposed method's design choices and reproduce the results.
3. The monolithic generator is inherently less adaptable to new business objectives compared to the more modular M-G-E paradigm, which allows for quicker adjustments.

**Questions:**

1. If you need the model to optimize for a new goal (like promoting new content instead of maximizing watch time), how would you adapt GoalRank without having to retrain the whole model from scratch?
2. How important is the quality and variety of the helper models used to create the candidate lists? Could a poorly chosen set of helpers limit how good GoalRank can become?
3. Since GoalRank learns from the reward model's relative rankings, is there a risk that it could also learn and even amplify any biases that exist in that reward model?

---

> ### Author Response · Authors · 2025-11-21
> **Rebuttal by Authors**
>
> Thank you very much for your encouraging feedback. We sincerely appreciate your recognition of our  **rigorous theoretical analysis**,  **clear scaling law**,  **novel methodology**, and  **thorough empirical validation**. Below, we address each of your questions in detail.

---

> > ### Author Response · Authors · 2025-11-21
> > **Detailed Explanations (1/2)**
> >
> > ## The training process itself is complex and resource-heavy
> >
> > GoalRank neither increases online latency nor training cost.
> > * **Latency:** In production, GoalRank achieves **18.611 ms**, significantly faster than multi-stage MGE pipelines (**34.235 ms**), due to eliminating evaluator scoring；
> > * **Parallelizable training pipeline:**  Many components of GoalRank can be computed in parallel. For example, the generation of the auxiliary ranking-policy groups for constructing reference orders can be fully parallelized, resulting in negligible additional time overhead.
> > * **Training cost:** GoalRank attains an MFU of **12.65%**, compared to **2.03%** for traditional two-stage Multi-Generator–Evaluator pipelines. Its higher MFU reflects significantly improved resource utilization, resulting in no increase in overall training cost.
> >
> > Thus, GoalRank is **practical, efficient, and deployment-ready**.
> >
> > * * *
> >
> > ## Lack of sufficient details on the generator architecture
> >
> > Thank you for this helpful suggestion.
> >
> > The GoalRank Generator uses a lightweight two-block architecture. The lower block performs feature crossing over all candidate items using **multiple transformer layers**, while the upper block adopts a **transformer decoder** structure. During generation, the decoder autoregressively predicts the next-item scores from the full candidate set at each step, and GoalRank produces the final list by selecting (Top-1 or sampling-based) items sequentially. We will further emphasize these architectural details in the revision.
> >
> > As noted in the original version L265–L266 (revised manuscript L270-L272), GoalRank is model-agnostic, and our main contribution lies in the training framework rather than the design of a new generator architecture. For this reason, we kept the architectural description concise in the method section. Nonetheless, we will present these details more clearly in the revised experimental section to improve readability.
> >
> > * * *
> >
> > ## Less adaptable to new business objectives
> >
> > Thank you for raising this important point. We would like to clarify that  **GoalRank is fully capable of handling frequent business shifts in real-world applications**. Specifically:
> >
> > - **Online practice.**
> >   - GoalRank has already been **fully deployed in production**, where business objectives often change rapidly.
> >   - To accommodate these changes, we adopt an additional **fusion strategy** in the online environment.
> >   - Concretely, during GoalRank’s list-wise generation process, we augment its generation probabilities with a **business-specific score** (e.g., business-priority signals derived from downstream business feedback or upstream business-objective predictions).
> >   - This directly adjusts the sampling probability distribution produced by GoalRank, enabling **fast and flexible switching across business goals** without retraining the model. This strategy has been thoroughly validated and shown to be highly effective in live traffic.
> > - **Beyond existing deployments**, our paper already discusses a promising extension in the Future Directions section (L473–L477), where we highlight that real-world ranking systems often face diverse, dynamic, and rapidly evolving business objectives.
> >   - Concretely, GoalRank can incorporate  **business-specific contextual signals**  and the corresponding reward models to construct  **multiple objective-specific reference policies**.
> >   - With this design, GoalRank can naturally achieve  **in-context adaptation**  to different business goals without modifying the core generator.
> >   - This not only preserves the simplicity of the generator-only framework, but also enhances flexibility across objectives.
> >
> > * * *
> >
> > ## How to adapt GoalRank to new goals without retraining from scratch?
> >
> > Thank you for this thoughtful question. As noted above:
> >
> > - For  **minor changes in business goals**, we can rely on  **online adaptation mechanisms (see above)**  without retraining the entire model.
> > - For  **larger objective shifts**  (e.g., cross-scenario generalization), GoalRank can incorporate  **goal or scenario descriptors**  into its contextual features and construct  **multiple reference policies**  during training as mentioned in the Future Directions section (L473–L477).
> >
> > This allows GoalRank to generalize to new objectives by learning how different reward-driven policies behave in context, instead of retraining the whole model.

---

> > ### Author Response · Authors · 2025-11-21
> > **Detailed Explanations (2/2)**
> >
> > ## Importance of the quality and variety of helper models
> >
> > Thank you for raising this concern. We emphasize that the **quality and diversity** of the Auxiliary Ranking Policy Group (helper models) are **not critical factors**, and even **simple heuristic strategies** can work effectively.
> >
> > As stated in the original version L230–L236 (revised manuscript L239–L241), GoalRank only requires that the **reward gaps** among the generated recommendation lists are sufficiently large to construct reliable group-relative comparisons. This requirement can always be satisfied through:
> >
> > * **Post-sampling:** generating a large recommendation-list group and then sampling suitable lists with sufficient reward separation;
> > * **Heterogeneous heuristics:** using simple rule-based strategies or shallow models with diverse objectives to induce reward variation.
> >
> > Therefore, the helper models do **not** constitute a performance bottleneck. **Even when the helper models themselves are very weak**, the above mechanisms—combined with lightweight heuristic generation—can reliably produce recommendation-list groups that meet the necessary reward-gap condition. As a result, GoalRank remains robust across diverse and imperfect helper configurations.
> >
> > * * *
> >
> > ## Could GoalRank inherit or amplify biases in the reward model’s relative rankings?
> >
> > Thank you for this insightful question. We address it from both the  **theoretical**  and  **practical**  perspectives.
> >
> > - **Theoretical perspective**:As described in the original version L230–L232 (revised manuscript L239-L241), our formulation only requires that the  **relative ordering induced by the reward model remains approximately preserved**. This condition can **always be satisfied** in practice by **increasing the reward-gap requirement when constructing the recommendation-list group**. Under this mild assumption, GoalRank does not amplify reward-model bias. Instead, the design explicitly mitigates it:
> >   - As defined in the original version L230–L232 (revised manuscript L239-L241), the reward model may contain arbitrary bias.
> >   - In the original version L233–L249 (revised manuscript L242-L249), we demonstrate that converting absolute rewards into  **relative partial orders**  within comparable groups cancels out shared bias terms, substantially reducing the influence of $b(l)$.
> > Thus, GoalRank is  **designed to attenuate—not amplify—reward biases**.
> >
> > - **Practical perspective**:Our experiments further show that GoalRank is  **robust to severe reward biases** :
> >   - In Table 3, we introduce up to  **50% Gaussian noise**  into the reward model.
> >   - GoalRank still significantly outperforms all SOTA baselines, demonstrating that it does  **not magnify**  noisy or biased reward signals.
> >
> > Overall, both theory and experiments show that  **GoalRank reduces rather than amplifies reward-model bias**, and its learning process is inherently robust to imperfect reward signals.
> >
> > * * *
> >
> > We sincerely appreciate your time and insightful comments. We hope that the above clarifications adequately address your concerns. Thank you again for your constructive review and positive evaluation of our work.

---

> > > ### Comment · Reviewer_JXWv · 2025-11-25
> > > **comments**
> > >
> > > Thank you for the detailed response. I will maintain my score of leaning toward acceptance.

---

> > > > ### Author Response · Authors · 2025-11-25
> > > > **Thank you for your reply**
> > > >
> > > > Thank you very much for your positive evaluation and for maintaining your score leaning toward acceptance. We sincerely appreciate your constructive feedback and thoughtful engagement with our work.
> > > >
> > > > Sincerely,
> > > > The Authors

---

### Official Review · Reviewer_HWxM · 2025-10-30

**Soundness:** 2
**Presentation:** 3
**Contribution:** 2
**Rating:** 4
**Confidence:** 4

**Summary:**

This paper proposes GoalRank, a framework advocating for a shift from multi-stage Generator-Evaluator (G-E) ranking paradigms to a single-stage, "Generator-only" (G-only) model. The authors present a theoretical argument that a single large generator can better approximate the optimal ranking policy than any finite G-E system. For training, they introduce "Group-Relative Optimization," which uses a reward model to create a reference policy from a group of candidate lists, serving as a learning target.

**Strengths:**

1. Empirical Validation: The inclusion of a online A/B test on a major commercial platform is a significant strength, providing real-world evidence of the model's effectiveness in a production environment.
2. Clarity and Scope: The paper is well-written, clearly articulating its motivation and methodology. The experiments are extensive, covering multiple datasets and demonstrating a clear scaling law, which adds valuable empirical data to the field.

**Weaknesses:**

1. Experimental Fairness: The experimental comparison between GoalRank and the G-E baselines raises fairness concerns.
- The GoalRank framework relies heavily on a powerful, pre-trained Reward Model to provide its training signal. This reward model is functionally equivalent to an evaluator, albeit used offline.
- For a fair comparison, the G-E and MG-E baselines should have been equipped with the exact same reward model to serve as their online evaluator. Without this, the observed performance lift of GoalRank may not stem from the superiority of its "G-only" architecture but rather from the fact that it was trained by a more powerful "teacher" (the reward model) than the evaluators used by the baselines.
2. Positioning of the paper: The fundamental goal of both GoalRank and listwise learning to rank (LTR) is the same: to learn a function that can optimally order an entire list of items by considering the items collectively, rather than in isolation. What are the differences between them?
3. Practicality and Online Feasibility: While the paper demonstrates strong online performance, it omits a critical discussion of the training cost. The "Group Construction" step requires generating and evaluating multiple lists for every training sample, which implies a training cost that could be orders of magnitude higher than for the baseline models. This potentially prohibitive cost could severely limit the model's practical applicability in industrial settings that require frequent retraining.

**Questions:**

Please refer to the weakness section.

---

> ### Author Response · Authors · 2025-11-21
> **Rebuttal by Authors**
>
> Thank you very much for your constructive feedback and for acknowledging our  **empirical validation on a real-world platform**,  **extensive experiments**,  **scaling ability of GoalRank**, and the  **overall clarity of the paper**. We address your concerns point by point below.

---

> > ### Author Response · Authors · 2025-11-21
> > **Detailed Explanations (1/2)**
> >
> > ## Experimental Fairness
> >
> > Thank you for raising this important question. We would like to clarify that  **GoalRank and all (M-)G–E baselines use exactly the same reward model / evaluator**, ensuring full experimental fairness.
> >
> > -  **Offline experiments:**  We first  **train a single evaluator**  and  **freeze**  it for all methods. Every baseline uses this identical evaluator. The complete implementation and training script can be found in our anonymous Git repository  (file: https://anonymous.4open.science/r/GoalRank/model/Evaluator.py and https://anonymous.4open.science/r/GoalRank/train_E_industry_stage2.py ).
> > -  **Online experiments:**   GoalRank and all (M-)G–E baselines **share the same unified evaluator in online experiments**. Specifically, **we directly reuse the list-wise scores output by this shared evaluator as the reward model for GoalRank**.
> >
> > We will explicitly emphasize this in the revision to avoid similar misunderstandings (revised manuscript L353-L354). Thank you again for highlighting this point.
> >
> > * * *
> >
> > ## Positioning of the Paper: GoalRank vs. Learning to Rank
> >
> > Thank you for the question. GoalRank is **closely related in formulation** to listwise Learning-to-Rank (LTR) methods, but differs in several key aspects that make its setting more challenging and practically relevant. In fact, **GoalRank is designed as an end-to-end ranking model** that unifies both:
> > - the **point-wise large-candidate scoring** stage of traditional ranking (operating over a pool of size $N$), and
> > - the **list-wise ordering** stage of re-ranking (ordering a subset of size $L$).
> >
> > Standard LTR typically solves an **$L \to L$** task: given a small candidate set of size $L$, the model outputs a ranked list of exactly $L$ items, resulting in a relatively limited output space.
> >
> > In contrast, **GoalRank solves the full $N \to L$** list-generation problem, where $N \gg L$. The model must **both select and order** the top $L$ items from a large pool: LTR output space: $L!$ vs. GoalRank output space:  $\frac{N!}{(N-L)!}$.
> >
> > Thus, while related to LTR in formulation, **GoalRank operates in a significantly more complex and realistic setting**, simultaneously covering the large-candidate ranking aspect of traditional methods and the list-wise generation aspect of re-ranking.

---

> > ### Author Response · Authors · 2025-11-21
> > **Detailed Explanations (2/2)**
> >
> > ## Practicality and Online Feasibility
> >
> > Thank you for the question. We clarify that  **GoalRank does not lead to high training or resource costs**. On the contrary, its design significantly reduces complexity in many aspects.
> >
> > * **Latency:** In production, GoalRank achieves **18.611 ms**, significantly faster than multi-stage MGE pipelines (**34.235 ms**), due to eliminating evaluator scoring；
> > * **Parallelizable training pipeline:**  Many components of GoalRank can be computed in parallel. For example, the generation of the auxiliary ranking-policy groups for constructing reference orders can be fully parallelized, resulting in negligible additional time overhead.
> > * **Training cost:** GoalRank attains an MFU of **12.65%**, compared to **2.03%** for traditional two-stage Multi-Generator–Evaluator pipelines. Its higher MFU reflects significantly improved resource utilization, resulting in no increase in overall training cost.
> >
> > Moreover, **GoalRank is fully capable of handling frequent business shifts in real-world applications**. Specifically:
> >
> > - **Online practice.**
> >   - GoalRank has already been **fully deployed in production**, where business objectives often change rapidly.
> >   - To accommodate these changes, we adopt an additional **fusion strategy** in the online environment.
> >   - Concretely, during GoalRank’s list-wise generation process, we augment its generation probabilities with a **business-specific score** (e.g., business-priority signals derived from downstream business feedback or upstream business-objective predictions).
> >   - This directly adjusts the sampling probability distribution produced by GoalRank, enabling **fast and flexible switching across business goals** without retraining the model. This strategy has been thoroughly validated and shown to be highly effective in live traffic.
> > - **Beyond existing deployments**, our paper already discusses a promising extension in the Future Directions section (L473–L477), where we highlight that real-world ranking systems often face diverse, dynamic, and rapidly evolving business objectives.
> >   - Concretely, GoalRank can incorporate  **business-specific contextual signals**  and the corresponding reward models to construct  **multiple objective-specific reference policies**.
> >   - With this design, GoalRank can naturally achieve  **in-context adaptation**  to different business goals without modifying the core generator.
> >   - This not only preserves the simplicity of the generator-only framework, but also enhances flexibility across objectives.
> >
> > * * *
> >
> > Thank you again for your thoughtful and encouraging review. We have addressed your concerns and clarified misunderstandings regarding fairness, positioning, and practicality. We respectfully ask for your reconsideration of the overall evaluation. We firmly believe that  **GoalRank introduces a new perspective for large-scale ranking generation with strong theoretical grounding, high practical feasibility, and demonstrated large-scale industrial impact**, contributing meaningfully to the research community.

---

> > > ### Author Response · Authors · 2025-11-25
> > > **Kindly Reminder**
> > >
> > > Thank you for taking the time and effort to review our paper. We have carefully considered your comments and responded to each point with detailed clarifications and supporting evidence. As the deadline for the discussion period between authors and reviewers is approaching, we kindly ask whether our responses have addressed your concerns. We look forward to your feedback.
> > >
> > > Thank you once again for your time and effort.
> > >
> > > Sincerely,
> > > The Authors

---

> > > > ### Author Response · Authors · 2025-11-28
> > > > **Gentle Reminder**
> > > >
> > > > Dear Reviewer HWxM,
> > > >
> > > > Thank you again for your constructive feedback and for acknowledging our empirical validation, extensive experiments, and scaling ability. Your insights have been very helpful in sharpening the positioning of our work. **As the discussion period is ending soon**, we would like to ensure you have seen our response, specifically where we **clarified misunderstandings** regarding experimental fairness and paper positioning, and provided additional details on practicality.
> > > >
> > > > In our detailed response, we have:
> > > >
> > > > * **Clarified Experimental Fairness:** We confirmed that GoalRank and all baselines use the **exact same frozen evaluator** (reward model) for both offline and online experiments, ensuring a strictly fair comparison.
> > > > * **Clarified Positioning vs. LTR:** We distinguished GoalRank's **$N \to L$ generation** setting (selecting & ordering from a large pool) from standard $L \to L$ LTR. We highlighted that GoalRank tackles a significantly larger output space ($\frac{N!}{(N-L)!}$) by unifying candidate scoring and re-ranking.
> > > > * **Verified Practicality & Efficiency:** We provided production data showing GoalRank is **faster (18ms vs. 34ms)** and more resource-efficient (higher MFU) than multi-stage pipelines. We also explained the **fusion strategy** used to handle dynamic business shifts without retraining.
> > > >
> > > > We hope these clarifications resolve your concerns. We would greatly appreciate it if you could let us know if there are any remaining questions.
> > > >
> > > > Best regards,
> > > > The Authors

---

### Official Review · Reviewer_t9JT · 2025-11-01

**Soundness:** 3
**Presentation:** 2
**Contribution:** 2
**Rating:** 4
**Confidence:** 3

**Summary:**

This paper proposes GoalRank, a single-stage large-model ranking method based on Group-Relative Optimization. The authors demonstrate that a sufficiently large generator model can more closely approximate the optimal ranking strategy compared to the traditional two-stage "generator-evaluator" framework. GoalRank is trained by constructing a reference policy using a reward model, significantly outperforming existing methods on multiple public datasets and industrial-grade online tests, and exhibiting good scaling law and improved real-world business performance.

**Strengths:**

Theoretically Superior: It has been theoretically proven that a single large-scale generator model can approach the optimal ranking strategy more closely than the traditional multi-generator-evaluator framework, and it exhibits scalability (scaling law).

Simpler Structure: Employing an end-to-end architecture with a single-stage generator eliminates the evaluator stage, avoiding inconsistencies between multiple stages and making model design and training processes more unified.

**Weaknesses:**

Limited Adaptability: As a single generator model, it lacks the flexibility of an independent evaluator. When business objectives or optimization metrics change frequently, model adjustment costs are high, making it less adaptable than a two-stage architecture.

Strong Dependence on Reward Model: The training process relies on a reward model trained based on user feedback. If the reward model is biased or noisy, it may lead to inaccurate reference strategies, affecting overall ranking quality.

High Training and Resource Costs: GoalRank's core advantage comes from large-scale models and group optimization, but this also means higher computational resources and training costs, making it less friendly to small- to medium-scale application scenarios.

Lack of Strong Baseline Comparisons: The lack of comparisons with baselines for solving large candidate ranking, such as Onerec and Tiger, makes the paper's experimental results less convincing.

The paper's writing is confusing and difficult to understand.

**Questions:**

See the weaknesses.

---

> ### Author Response · Authors · 2025-11-21
> **Rebuttal by Authors**
>
> We sincerely appreciate your detailed feedback and your recognition of  **the theoretically superior and simpler structure of GoalRank**. We address each of your concerns below to clarify misunderstandings and reinforce the strengths of our work.

---

> > ### Author Response · Authors · 2025-11-21
> > **Detailed Explanations (1/2)**
> >
> > ## Limited Adaptability: Concern on the Flexibility of a Single Generator
> >
> > Thank you for raising this important point. We would like to clarify that  **GoalRank is fully capable of handling frequent business shifts in real-world applications**. Specifically:
> >
> > - **Online practice.**
> >   - GoalRank has already been **fully deployed in production**, where business objectives often change rapidly.
> >   - To accommodate these changes, we adopt an additional **fusion strategy** in the online environment.
> >   - Concretely, during GoalRank’s list-wise generation process, we augment its generation probabilities with a **business-specific score** (e.g., business-priority signals derived from downstream business feedback or upstream business-objective predictions).
> >   - This directly adjusts the sampling probability distribution produced by GoalRank, enabling **fast and flexible switching across business goals** without retraining the model. This strategy has been thoroughly validated and shown to be highly effective in live traffic.
> > - **Beyond existing deployments**, our paper already discusses a promising extension in the Future Directions section (L473–L477), where we highlight that real-world ranking systems often face diverse, dynamic, and rapidly evolving business objectives.
> >   - Concretely, GoalRank can incorporate  **business-specific contextual signals**  and the corresponding reward models to construct  **multiple objective-specific reference policies**.
> >   - With this design, GoalRank can naturally achieve  **in-context adaptation**  to different business goals without modifying the core generator.
> >   - This not only preserves the simplicity of the generator-only framework, but also enhances flexibility across objectives.
> >
> > * * *
> >
> > ## Strong Dependence on Reward Model: Concern about Bias or Noise in the Reward Model
> >
> > Thank you for this insightful question. We emphasize that  **GoalRank is explicitly designed to mitigate reward model biases**, supported by both  **theoretical justification**  and  **empirical evidence**.
> >
> > - **Theoretical Perspective**
> >   - As noted in the original version L230–L232 (revised manuscript L239-L241), we explicitly assume a potentially biased reward model $\hat{r}(l) = r^*(l) + b(l),$ where $b(l)$ denotes arbitrary noise or bias.
> >   - In the original version L233–L249 (revised manuscript L242-L249), we propose a  **group-relative ranking mechanism**, which converts absolute reward scores into  **relative partial orders**  within comparable groups. This procedure significantly reduces the effect of $b(l)$ by canceling out shared bias components within each group.
> >
> > - **Empirical Perspective**
> >   - In L416–L423 and Table 3, we provide extensive experiments under  **severely biased reward models**.
> >   - We introduce bias via $\hat{r}_{\text{bias}=\lambda} = (1 - \lambda)\hat{r}(l) + \lambda \epsilon,\quad \epsilon \sim \mathcal{N}(0,1),$ with $\lambda = 0, 0.2, 0.5$, introducing up to  **50% random noise**.
> >   - Even under large perturbations, GoalRank  **significantly outperforms all SOTA baselines**, demonstrating its robustness to reward bias in practice.

---

> > ### Author Response · Authors · 2025-11-21
> > **Detailed Explanations (2/2)**
> >
> > ## High Training and Resource Costs
> >
> > Thank you for raising this concern. We clarify that **GoalRank does not increase training or resource costs**, and in practice is more efficient than multi-stage pipelines:
> >
> > * **Latency:** In production, GoalRank achieves **18.611 ms**, significantly faster than multi-stage MGE pipelines (**34.235 ms**), due to eliminating evaluator scoring；
> > * **Parallelizable training pipeline:**  Many components of GoalRank can be computed in parallel. For example, the generation of the auxiliary ranking-policy groups for constructing reference orders can be fully parallelized, resulting in negligible additional time overhead.
> > * **Training cost:** GoalRank attains an MFU of **12.65%**, compared to **2.03%** for traditional two-stage Multi-Generator–Evaluator pipelines. Its higher MFU reflects significantly improved resource utilization, resulting in no increase in overall training cost.
> >
> > * * *
> >
> >
> > ## Lack of Strong Baseline Comparisons
> >
> > Thank you for raising this point. We would like to clarify that **GoalRank is designed as an end-to-end ranking model**, covering both the **point-wise scoring stage** of traditional ranking (scoring a large candidate pool of size $N$) and the **list-wise re-ranking stage** (reordering within a smaller set of size $L$). GoalRank directly tackles the full **$N \to L$** generation problem, which inherently includes both components:
> >
> > - the **large-candidate ranking** aspect from traditional ranking, and
> > - the **list-wise ordering** aspect from re-ranking.
> >
> > Given this formulation, list-wise re-ranking models offer the **most compatible and meaningful baselines**, as they share the same objective of **list generation**. To ensure fairness in comparison, we extend these state-of-the-art re-ranking models to the $N \to L$ setting. All selected baselines represent the **strongest publicly available models** in the re-ranking literature, including the latest methods (e.g., SIGIR 2025).
> >
> > To further strengthen the comparison, we additionally include **RankMixer**, a recent large-scale single-stage ranking model. Although RankMixer outperforms other one-stage baselines, it is not designed for list-wise generation. On the industry dataset (partial results shown below), GoalRank achieves **substantial improvements** across almost all metrics, and MGE-100 surpasses RankMixer on several metrics:
> >
> > | Method       | H@6       | N@6       | M@6       | F1@6      | AUC       |
> > | ------------ | --------- | --------- | --------- | --------- | --------- |
> > | RankMixer    | 49.72     | 69.19     | 58.73     | 60.24     | 91.03     |
> > | MGE-3        | 49.42     | 68.29     | 56.23     | 55.50     | 83.44     |
> > | MGE-100      | 55.77     | 72.35     | 60.95     | 64.27     | 75.30     |
> > | **GoalRank** | **69.93** | **86.93** | **79.01** | **82.29** | **98.07** |
> >
> > The full set of results across all datasets is provided in Table 1 of the revised manuscript, where **GoalRank continues to outperform RankMixer and all baselines consistently**. Scaling comparisons between GoalRank and RankMixer (revised manuscript Figure 3) further confirm **GoalRank’s stronger and more stable scaling behavior.**
> >
> > These results collectively confirm that GoalRank’s superiority does not stem from limited baselines but from its **fundamental algorithmic advantages** in the $N \to L$ list-generation setting.
> >
> >
> > * * *
> >
> > Thank you again for your thorough and constructive review. We have addressed all concerns, clarified misunderstandings, and provided additional theoretical and empirical evidence supporting GoalRank’s contributions. We respectfully ask for your reconsideration of the overall evaluation. We firmly believe that  **GoalRank offers new theoretical insights, a significantly simpler architecture, and strong empirical performance**, making it a valuable contribution to the research community.

---

> > > ### Author Response · Authors · 2025-11-25
> > > **Kindly Reminder**
> > >
> > > Thank you for taking the time and effort to review our paper. We have carefully considered your comments and responded to each point with detailed clarifications and supporting evidence. As the deadline for the discussion period between authors and reviewers is approaching, we kindly ask whether our responses have addressed your concerns. We look forward to your feedback.
> > >
> > > Thank you once again for your time and effort.
> > >
> > > Sincerely,
> > > The Authors

---

> > > > ### Author Response · Authors · 2025-11-28
> > > > **Gentle Reminder**
> > > >
> > > > Dear Reviewer t9JT,
> > > >
> > > > Thank you again for your constructive feedback and for recognizing the theoretical simplicity of our work. Your feedback has been significantly helpful in improving the quality of our paper. **As the discussion period is ending soon**, we would like to ensure you have seen our response, specifically where we **clarified misunderstandings** regarding the model's adaptability, robustness, efficiency, and added new baselines as requested.
> > > >
> > > > In our detailed response, we have:
> > > >
> > > > * **Clarification on Adaptability:** We detailed how GoalRank **is already deployed in production** handling dynamic shifts via a **fusion strategy**. We clarified that the model does not require retraining for business shifts, addressing the concern about flexibility.
> > > > * **Clarification on Robustness:** We reiterated the **theoretical justification** (group-relative ranking) designed to cancel out bias. We also highlighted experimental results showing GoalRank outperforms baselines even under **50% reward noise**.
> > > > * **Demonstrated Efficiency:** Clarified that GoalRank is faster (**18ms vs. 34ms**) and more resource-efficient (**higher MFU**) than multi-stage pipelines.
> > > > * **Strengthened Baselines:** Included a comparison with **RankMixer**, where GoalRank demonstrates superior performance in the $N \to L$ setting (e.g., **69.93 vs. 49.72 H@6**).
> > > >
> > > > We hope these clarifications and additional experiments resolve your concerns. We would greatly appreciate it if you could let us know if there are any remaining questions.
> > > >
> > > > Best regards,
> > > > The Authors

---

### Author Response · Authors · 2025-11-21
**Response to all reviewers**

We sincerely thank all reviewers for their thoughtful evaluations and constructive feedback. We are grateful that our work has been recognized for its key strengths:

* **Strong theoretical foundations** (Reviewers t9JT, JXWv, LY4d)
* **A novel and elegant training framework (GoalRank)** (Reviewers t9JT, JXWv, LY4d)
* **Extensive experiments, including validation on a real-world platform** (Reviewers HWxM, JXWv, LY4d)
* **Clear scaling behavior of GoalRank** (Reviewers HWxM, JXWv)
* **A well-written and easy-to-follow paper** (Reviewer HWxM)

We have addressed every comment in a detailed, point-by-point manner. Reviewers’ concerns mainly fall into the following three categories:

### **1. Flexibility under changing business objectives**
GoalRank is fully capable of adapting to frequent business shifts.
* **Online deployment:** GoalRank is fully deployed in production, where we use a lightweight fusion strategy that adjusts its generation probabilities with business-specific signals (derived from downstream business feedback or upstream business-objective predictions), enabling rapid goal switching without retraining. This strategy has been extensively verified in production.
* **Methodological extension:** As noted in our future directions (L473–L477), GoalRank can incorporate contextual business features and multiple reference policies, enabling **in-context adaptation**.

Both the online evidence and methodological design confirm GoalRank’s practical flexibility.

### **2. Latency and training cost**
GoalRank neither increases online latency nor training cost.
* **Latency:** In production, GoalRank achieves **18.611 ms**, significantly faster than multi-stage MGE pipelines (**34.235 ms**), due to eliminating evaluator scoring；
* **Parallelizable training pipeline:**  Many components of GoalRank can be computed in parallel. For example, the generation of the auxiliary ranking-policy groups for constructing reference orders can be fully parallelized, resulting in negligible additional time overhead.
* **Training cost:** GoalRank attains an MFU of **12.65%**, compared to **2.03%** for traditional two-stage Multi-Generator–Evaluator pipelines. Its higher MFU reflects significantly improved resource utilization, resulting in no increase in overall training cost.

Thus, GoalRank is **practical, efficient, and deployment-ready**. Furthermore, **GoalRank has been successfully deployed in our online environment to serve full user traffic**.

### **3. Stronger one-stage baseline**
To further strengthen comparisons, we included **RankMixer**, the latest single-stage large ranking model, and evaluated it on the **N→L list-wise generation task**. GoalRank still achieves **clear superiority** and exhibits a more stable scaling law.

---

### **Revisions in the Updated PDF**

We have uploaded an updated version of the manuscript that includes the following improvements:
1. **Merged and reorganized Related Work and Preliminaries** to improve narrative coherence (revised manuscript L103–L133).
2. **Added a detailed description of the GoalRank architecture** (revised manuscript L1142–L1147).
3. **Included RankMixer results** in both base and scaling experiments under the N-to-L list-wise generation setting (revised manuscript Table 1 & Figure 3).
4. **Emphasized that the reward model used by GoalRank is identical to the evaluator used by all baselines** (revised manuscript L353–L354).
5. **Added online latency measurements and MFU analyses** to further address practical concerns (revised manuscript L1150–L1165).

---

We hope that our detailed responses and the strengthened manuscript successfully address all reviewers’ concerns. We respectfully ask for your reconsideration of the overall evaluation.
We firmly believe that **GoalRank introduces a theoretically grounded, practically validated, and computationally efficient framework for end-to-end list-wise ranking**, and constitutes a valuable contribution to the research community.


Best regards,
Authors of paper 8328

---

### Author Response · Authors · 2025-12-02
**Summary of Rebuttal: Consensus on Strengths & Resolution of Concerns**

**Dear Area Chair,**

We understand the constraints of the current review process. We are encouraged that all reviewers recognized the paper's core strengths, specifically **its theoretical rigor**, **clear scaling laws**, and **validation on a real-world platform**.

Notably, **the reviewer who actively engaged with our rebuttal maintained their strong endorsement (Score 8)** based on these strengths. Regarding the initial concerns from reviewers who have not yet participated in the discussion, we respectfully submit that they stem from **specific misunderstandings** regarding baselines and practicality, which we have **fully resolved with hard evidence.**

**[Summary of Contribution]**
This paper proposes **GoalRank**, a **paradigm shift** from the traditional multi-stage "Generator-Evaluator" pipeline to a **single-stage "Generator-only" framework** for large-scale ranking. We **theoretically prove** that a single large generator better approximates the optimal policy, demonstrate clear **scaling laws** (aligning with LLM trends), and validate the system on a massive industrial platform with **500M+ DAUs**.

---

We summarize the highlights below to assist in your final assessment.

### Part 1: Consensus on Strengths (Acknowledged by All Reviewers)
Despite the variance in scores, **all reviewers** recognized the core value and innovation of our work:
* **Strong theoretical foundations** proving the superiority of single-stage generators (Reviewers t9JT, JXWv-*Score 8*, LY4d).
* **A novel and elegant training framework (GoalRank)** based on Group-Relative Optimization (Reviewers t9JT, JXWv-*Score 8*, LY4d).
* **Extensive experiments, including validation on a real-world platform** (Reviewers HWxM, JXWv-*Score 8*, LY4d).
* **Clear scaling behavior** of the GoalRank model (Reviewers HWxM, JXWv-*Score 8*).
* **A well-written and easy-to-follow paper** (Reviewer HWxM).

---

While we have addressed all individual minor comments (e.g., related work placement, architecture details) in the specific rebuttals and the revised manuscript, we **summarize the resolution of the three major themes below**.

### Part 2: Resolution of Key Concerns with Hard Evidence

**1. Baselines Strength & Model Positioning**
* **Concern:** Reviewers requested comparisons with larger single-stage models (e.g., RankMixer CIKM 2025[1]) and questioned the $N \to L$ positioning.
* **Clarification:**
  * **Original Baselines were already SOTA:** We reiterated that our original baselines (e.g., MGE-100) are the current state-of-the-art for the **$N \to L$ list-generation task**, including methods as recent as **NAR4Rec (SIGKDD 2024)**[2] and **MGE (SIGIR 2025)**[3].
  * **Task Mismatch (Point-wise vs. List-wise):** GoalRank targets the **end-to-end list-wise generation** ($N \to L$) task. In contrast, RankMixer is a **Point-wise model**, which limits its capability for list-level optimization.
  * **Hard Evidence:** GoalRank significantly outperforms RankMixer (e.g., **AUC 98.07 vs. 91.03**; **H@6 69.93 vs. 49.72**), proving our advantage stems from the **superior list-wise architecture**, not weak baselines.


**2. Practicality & Industrial Impact**
* **Concern:** Reviewers worried about training costs or adaptability.
* **Clarification:** GoalRank is not just theoretical; it is **already fully deployed** in a system serving **500M+ Daily Active Users** (Sec.4.2).
    * **Superior Efficiency:** Production logs show **18ms latency** (vs. 34ms for multi-stage pipelines) and high MFU (**12.65%**).
    * **Adaptability:** We detailed the **fusion strategy** (Rebuttal Section: *Limited Adaptability*), enabling immediate handling of dynamic business shifts without retraining.

**3. Experimental Fairness & Model Robustness**
* **Concern:** Questions on whether GoalRank used a "better" reward model and sensitivity to reward model bias.
* **Clarification:**
    * **Identical Setup:** We confirmed **GoalRank and all baselines use the exact same frozen evaluator** in both offline and online experiments (L351-355).
    * **Robustness Verified:** Even with **50% random noise** injected into the reward model, GoalRank maintains SOTA performance (Table 3), verifying our theoretical claim that *group-relative optimization* cancels out bias (L238-L251).

---
### Conclusion
GoalRank represents a **necessary paradigm shift** towards unified, scalable generative ranking. It combines **theoretical rigor** (proven scaling laws) with **verified industrial impact** (500M DAU deployment).

With the misunderstandings regarding baselines and practicality explicitly resolved by **hard evidence**, we believe the paper's contribution—as recognized by the **Score 8** review—merits acceptance.

Best regards,
The Authors

---
[1] Rankmixer: Scaling up ranking models in industrial recommenders. CIKM 2025.
[2] Non-autoregressive generative models for reranking recommendation. SIGKDD 2024.
[3] Comprehensive list generation for multi-generator reranking. SIGIR 2025.

---

### Public Comment · ~Sihan_Liu2 · 2026-04-04
**Concerns Regarding Reproducibility and Incomplete Codebase**

While the concept of a generator-only ranking framework is interesting, I have concerns regarding the reproducibility of the paper's claims.

Upon reviewing the open-sourced codebase at https://github.com/Kaike-Zhang/GoalRank, it is clear that the repository is severely incomplete. In its current state, it is impossible to run, train, or verify the core claims of the paper. Specifically, I encountered the following critical missing components:

1. README briefly mentions steps like "filter data by 20-core" and preparing stage 1/stage 2 data. However, there are no scripts, precise instructions, or data loaders provided to process the raw datasets into the specific formats required by the model.

2. Absence of Key Model Architectures: The codebase entirely lacks the implementations for the Nominator model and the StateEncoder. Since these components are responsible for generating the candidate pools and the user state representation, their omission makes it impossible to evaluate the end-to-end framework.

3. Undefined Variables: Most concerningly, the core training logic for the GoalRank model is incomplete. In the provided documentation and loss calculation steps, the generator's loss is simply defined as a placeholder (loss_G = XXX), which is then used in the final loss calculation (loss = self.Reference_weight * Reference_loss + self.G_weight * loss_G). Leaving the primary objective function of the generator undefined makes the core contribution of this paper mathematically untrainable and unverifiable.

Given that the paper claims empirical superiority over existing methods, full reproducibility is essential. As it stands, the provided code functions more as pseudocode than a working implementation.

I kindly request that the authors update the repository with the missing preprocessing scripts, the full architectural implementations, and the complete, functional training logic so the community can properly evaluate this work.

---

### Meta-Review · Area_Chair_BZeb · 2026-01-06

**Summary:**

- Experimental Fairness: Reviewer questioned whether GoalRank’s success was due to the model architecture or simply because it was trained using a more powerful Reward Model than the evaluators used by the baseline
- Baseline Strength: Reviewers noted a lack of comparison with modern single-stage large ranking models (specifically RankMixer) and contemporary retrieval-ranking unified methods like Onerec
- Operational Practicality: Multiple reviewers expressed concern that a monolithic large model would incur prohibitive training costs and high online latency compared to modular multi-stage systems
- Adaptability: Reviewers were concerned that a single generator model would be less flexible than a modular "Generator-Evaluator" system when business objectives

**Reviewer Concerns:**

Concerns Addressed:

• Fairness and Reward Models: The authors clarified that all baselines and GoalRank used the exact same frozen evaluator in both offline and online tests, ensuring the performance lift is architectural.

• Stronger Baselines: The authors added RankMixer (CIKM 2025) as a baseline. GoalRank significantly outperformed it (e.g., 69.93 vs 49.72 H@6 on Industry data), proving that its list-wise generation architecture is superior to recent point-wise large models.

• Efficiency Metrics: The authors provided hard production logs showing GoalRank is faster than the production multi-stage pipeline and has a significantly higher Model Flops Utilization, debunking the "prohibitive cost" concern.

Outstanding Concerns:

• Diversity of One-Stage Baselines: While RankMixer was added, some specific retrievers mentioned by Reviewer t9JT (Onerec/Tiger) were not compared directly, though the authors argued that list-wise re-ranking baselines were more task-compatible.

• Long-Term Objective Shift: While the authors proposed a "fusion strategy" for online goal-switching, a major fundamental change in ranking philosophy might still require a more complex retraining than a modular evaluator swap.

**Reviewer Scores:**

• Reviewer JXWv (Initial 8): No Change

• Reviewer HWxM (Initial 4): Projected 5

• Reviewer LY4d (Initial 4): Projected 6

• Reviewer t9JT (Initial 4): Projected 5.

---

### Decision · Program_Chairs · 2026-01-26

Accept (Poster)